# Towards Internet-Scale Training For Agents

## Abstract

The predominant approach for training web navigation agents is to gather human demonstrations for a set of popular websites and hand-written tasks, but it is becoming clear that human data is an inefficient resource. We develop a pipeline to facilitate internet-scale training for agents without laborious human annotations. In the first stage, an LLM annotates 150k sites with agentic tasks. In the next stage, LLM agents complete tasks and produce trajectories. In the final stage, an LLM filters trajectories by judging their success. Language models are powerful data curation tools, identifying harmful content with an accuracy of 97%, judging successful trajectories with an accuracy of 82.6%, and producing effective data. We train agents based on *Qwen 3 1.7B* that are competitive with frontier LLMs as web agents, while being smaller and faster. Our top agent reaches a success rate of 56.9%, outperforming the data collection policy *Qwen 3 235B*, a 235 times larger *Llama 4 Maverick*, and reaching 94.7% of the performance of *Gemini 2.5 Flash*. We will be releasing code, models and data to reproduce the entire pipeline.

## 1 Introduction

The predominant approach for training LLM web navigation agents is to collect human demonstrations for a set of manually curated websites and tasks (Deng et al., 2023; Zhou et al., 2024b; Putta et al., 2024; Koh et al., 2024a; Liu et al., 2024; Lù et al., 2024; Rawles et al., 2023). Human data can be laborious to collect, and becomes costly to scale as the breadth of skills that users require from language model agents grows. There are more than 300M sites on the western internet according to The Common Crawl Foundation (2025), and the range of sites that researchers have annotated represents a tiny fraction of the vast available data. And crucially, the existing human data is *static*. There is a growing need to automate pipelines for training the next generation of language model agents in a *dynamic internet-scale environment*. This paper addresses the core challenge of building this environment—reducing dependence on human annotations. We develop an automatic pipeline that aims to facilitate internet-scale training for agents, which we refer to as the InSTA pipeline.

The InSTA pipeline has three stages. In the first stage, we employ a language model task proposer that annotates 150k sites with live web navigation tasks for agents to perform. Existing works are limited to 200 popular websites (He et al., 2024; Lù et al., 2024; Rawles et al., 2023; Deng et al., 2023; Zhou et al., 2024c; Murty et al., 2025) that researchers have annotated manually, and a handful of synthetic websites (Zhou et al., 2024b; Koh et al., 2024a; Yao et al., 2023a). Our first goal in this paper is to improve coverage of real-world sites. To accomplish this, we cast a wide net. Starting from the top 1M sites on the internet ranked by popularity, our task proposer filters down to 150k sites that have safe content. Safety is critical when building autonomous agents, and our task proposer succeeds at detecting harmful content with an accuracy of 97%. Tasks are generated for sites marked as safe by the task proposer, and we run language model agents to complete the generated tasks. Agent progress is then fed back to the task proposer, which reviews trajectories and judge evaluations in order to assign a harder task based on the latest content of the website, closing the loop.

Scaling the task generation loop, we annotate 150k diverse sites with challenging agentic tasks, and release an official huggingface dataset: *data-for-agents/insta-150k-v2*. Motivated by the importance of internet data to progress in modern deep learning, our second goal in this paper is an internet-scale data flywheel for training LLM agents. We approach this by harnessing LLMs as data curation tools. After generating tasks, the pipeline employs pretrained LLMs as agents to complete tasks and produce trajectories, which are filtered by a judge to select the best data. Agents control a virtual web browser and produce reasoning traces that contain function calls to interact with and navigate live webpages.

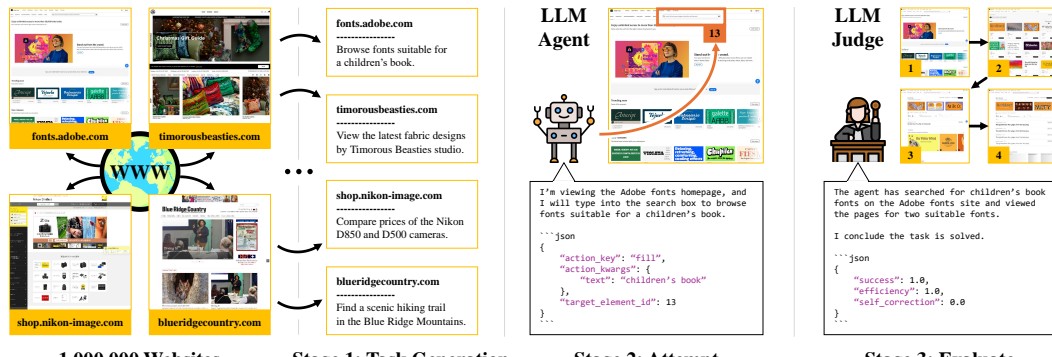

Figure 1: **Overview of the InSTA pipeline.** Our work unlocks a dynamic internet-scale environment that allows training small models to match frontier LLMs as agents, on a fraction of the budget. Starting from the top 1M sites on the internet, we annotate 150k sites with challenging agentic tasks, and release the entire pipeline, including code, models and an official huggingface dataset, on our website: anonymous-insta-pipeline.github.io.

The judge produces a reasoning trace that considers whether a trajectory is successful, and scores the agent on a continuous scale from 0 to 1. We scale the pipeline to create a large reasoning dataset for multimodal agents, with 2.2M screenshots, 2.2M traces for actions, and 150k traces for the judge.

Our data unlocks great potential in small language models as agents. We train a series of models based on *Qwen 3 1.7B* on varying scales of data from the InSTA pipeline, and match the performance of frontier LLM agents, on a fraction of the budget. Our top agent has a success rate of 56.9%, outperforming the data collection policy *Qwen 3 235B*, beating a 235 times larger *Llama 4 Maverick*, and reaching 94.7% of the performance of *Gemini 2.5 Flash*. To share our progress, we are releasing the entire pipeline, including code, models and data, on our website: anonymous-insta-pipeline.github.io.

## 2 RELATED WORKS

**Language Model Agents.**    There is an emerging paradigm in modern NLP using language models (Radford et al., 2019; Brown et al., 2020; Touvron et al., 2023a;b) as the backbone for agents (Andreas, 2022). These models show impressive reasoning capabilities (Bubeck et al., 2023; Zhong et al., 2024; Valmeekam et al., 2024) that allow them to generalize to downstream applications, such as web navigation, where observations differ from LLM training data. Search algorithms provide a secondary axis to improve the reasoning capabilities of language model agents (Yao et al., 2023b; Besta et al., 2024; Koh et al., 2024b; Zhou et al., 2024a) by providing an explicit algorithmic scaffold and allowing test-time compute to improve reasoning steps (Snell et al., 2024; Zhong et al., 2024). Although most recent work focuses on running language models as zero-shot agents, fine-tuning language models to improve their effectiveness as agents is becoming popular (Putta et al., 2024; Zeng et al., 2023; Zhang et al., 2023; Hong et al., 2023; Xie et al., 2024; Wang et al., 2024; Zhou et al., 2024c; Murty et al., 2025) as target benchmarks are becoming more difficult to solve zero-shot.

**Agent Pipelines.**    There are a growing number of agent pipelines aimed at fine-tuning language models to improve their effectiveness as agents (Mitra et al., 2024; Zeng et al., 2023; Putta et al., 2024; Chen et al., 2023; Ou et al., 2024; Zhou et al., 2024c; Murty et al., 2025). However, driven by the limited data available, many such works train on data with significant overlap with their test environment, either with different tasks for the same environment configuration as the test setting (Deng et al., 2023; Zhou et al., 2024c; Murty et al., 2025), or even the same tasks (Putta et al., 2024). We consider a setting where tasks and environment configurations (i.e. websites) are entirely separate between training and testing, creating a strong train-test split. This presents a challenge: human data for training LLM agents is limited (Deng et al., 2023; Lù et al., 2024). We address this challenge by reducing dependence on human data in agent pipelines. We employ LLMs as data curation tools to automatically design challenging tasks, and select the best training data. Our pipeline allows us to train small models that match frontier LLMs on Web Voyager (He et al., 2024), without using any data from Web Voyager. Contrast this with methods that train primarily on data from Web Voyager (Zhou et al., 2024c; Murty et al., 2025), and may not transfer to other benchmarks (Xue et al., 2025).

**Agent Datasets.** Datasets for training web navigation agents typically rely on human annotators to create tasks (Zhou et al., 2024b; Koh et al., 2024a; Rawles et al., 2023), and provide demonstrations for tasks (Deng et al., 2023; Lù et al., 2024; Rawles et al., 2023; Shen et al., 2024). However, the amount of data researchers have annotated represents a tiny fraction of the available internet data. There are more than 300M sites on the internet according to The Common Crawl Foundation (2025), yet existing datasets are limited to 200 popular sites that human annotators are familiar with (Deng et al., 2023; Lù et al., 2024; Shen et al., 2024). Human data can be laborious to collect, and becomes costly to scale as the capabilities users require from agents grows. And crucially, human data is *static*. Our work moves away from fixed datasets for training agents, and towards a dynamic internet-scale environment that grows with an ever-changing internet. We are not the first to build an environment (Zhou et al., 2024b; Koh et al., 2024a; Yao et al., 2023a; He et al., 2024), nor are we the first to consider synthetic data (Gandhi et al., 2024; Ou et al., 2024; Setlur et al., 2024; Tajwar et al., 2024), but we have solved a key challenge that unlocks the internet as the largest environment for agents.

**Language Model Judges.** Using LLMs to judge the correctness of responses is becoming popular to refine LLM predictions (Li et al., 2024), including to verify reasoning steps (Zhang et al., 2024), rejection sample (Snell et al., 2024; Sun et al., 2024), prioritize frontier nodes in search algorithms (Zhou et al., 2024a; Koh et al., 2024b), filter out harmful responses (Inan et al., 2023), provide feedback for response improvement (Madaan et al., 2023; Paul et al., 2024; Patel et al., 2024; Yuksekgonul et al., 2024), and provide ratings for alignment (Lee et al., 2024; Ouyang et al., 2024). Our use of language models to evaluate agents is inspired by Generative Verifiers (Zhang et al., 2024), and the multimodal verifier in He et al. (2024). One difference is our verifier predicts a reasoning trace that scores the agent from 0 to 1, which helps us rank trajectories to select the best data.

# 3 LANGUAGE MODEL AGENTS

Language model agents are a class of decision-making agents represented by $\pi_{\text{LLM}}(\mathbf{a}_t|\mathbf{s}_t, \mathbf{c}_n)$, a policy that processes multimodal observations $\mathbf{s}_t$ (from a virtual web browser in our case) and predicts textual actions $\mathbf{a}_t$ to complete a task $\mathbf{c}_n$. Underneath this abstraction, a large language model (LLM) generates actions via the next-token prediction, conditioned on a system prompt $\mathbf{x}_{\text{agent}}$.

$$\mathbf{a}_t = f^{\text{text}\rightarrow\text{act}}(\text{LLM}(\mathbf{x}_{\text{agent}}, \mathbf{c}_n, \text{Enc}(\mathbf{s}_t)))  \quad (1)$$

Environment representations for observations and actions typically differ from the expected input format of the language model (typically images and text), and functions are introduced that map the observations to a multimodal prompt $\text{Enc}(\cdot)$, and parse actions from the language model generated response $f^{\text{text}\rightarrow\text{act}}(\cdot)$. For web navigation, the environment state $\mathbf{s}_t$ is HTML DOM, and is often formatted as raw HTML code, an Accessibility Tree, Set-of-marks, or screenshots (Zhou et al., 2024b; Koh et al., 2024a; Chezelles et al., 2024; Shen et al., 2024). We built a fast Markdown parser that converts webpage observations into a compact readable format (refer to the code). Action formats vary between works, and we build on Schick et al. (2023)'s function-calling framework, where a language model generates code that is parsed into a function name and corresponding arguments. Given sets of function and argument names $L_i$, and sets of argument values $G_i$, the action space $\mathcal{A}$ is:

$$\mathcal{A} = L_{\text{func}} \times (L_{\text{arg1}} \times G_{\text{arg1}}) \times (L_{\text{arg2}} \times G_{\text{arg2}}) \times \cdots \times (L_{\text{argN}} \times G_{\text{argN}})  \quad (2)$$

where $L_{\text{func}}$ is the set of function names on the page object in the Playwright API (Microsoft, 2024), and function arguments have a name and value ($L_{\text{arg1}} \times G_{\text{arg1}}$) corresponding to the Playwright API. We allow the agent access to call arbitrary functions in Playwright (Microsoft, 2024), a Microsoft-developed browser automation library that wraps a headless web browser. The agent's goal is to complete a web navigation task specified via a natural language instruction $\mathbf{c} \in L$, starting from an initial URL, and operating the browser via function calls to the Playwright API until the desired task is complete, after which the agent calls the `stop` function and provides a final response:

$$\mathbf{a}_{\text{stop}} = (\text{"stop"}, (\text{"response"}, \text{"the task has been completed."}))  \quad (3)$$

We prompt the agent to produce a reasoning trace of a desired length (ablated in Figure 10) that contains function call actions as JSON in a fenced code block. To parse actions from the response, we employ a regex template that matches the first JSON code block, and a JSON decoder $f^{\text{text}\rightarrow\text{act}}(\cdot)$

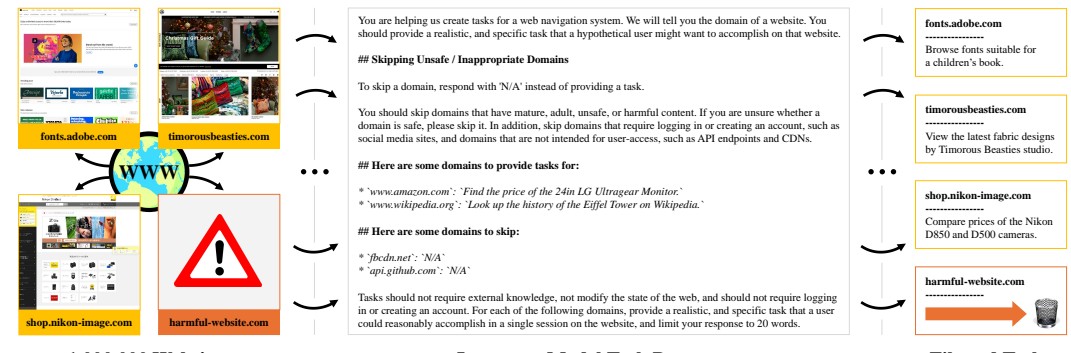

Figure 2: **Annotating 150k live sites with agentic tasks.** Starting from 1,000,000 websites, we employ a pretrained language model that marks sites as safe/unsafe for annotation, and assigns a realistic task that a hypothetical user might want to accomplish on each site. The task proposer aggressively filters out 85% of websites from the pipeline, resulting in 150k safe websites annotated with realistic tasks.

to parse the contents within the code block. When parsing fails due to invalid syntax, we allow the agent to generate a second response. Equipped with a language model agent that makes function calls with the Playwright API, we may consider the crucial task of obtaining large and diverse data.

# 4 INTERNET-SCALE TASK GENERATION

Training the next generation of LLM agents requires a large and diverse set of websites and tasks beyond what researchers have annotated so far (Deng et al., 2023; Zhou et al., 2024b; Koh et al., 2024a; Liu et al., 2024; Lù et al., 2024; Rawles et al., 2023; He et al., 2024). We develop an approach to efficiently annotate vast numbers of sites from diverse sections of the internet with agentic tasks. Our approach introduces two important desiderata: (1) it should not rely on human annotations, and (2) tasks should derive from a feedback process that deeply explores the environment.

## 4.1 LANGUAGE MODEL TASK PROPOSER

The key idea in stage one of the pipeline is a feedback loop, where a language model task proposer $\psi_{\text{LLM}}(\cdot)$ guides exploration on a website via an initial easy task. We then run a language model agent to explore the site, conditioned on the initial task, which produces an exploratory trajectory that is fed back to the task proposer. Conditioned on a trajectory that deeply explores the website, the task proposer then creates a harder, grounded task. This process is summarized as an equation.

$$\mathbf{c}_{n+1} \sim \psi_{\text{LLM}}(\cdot | \mathbf{w}, \mathbf{c}_n, \underbrace{\mathbf{s}_1, \mathbf{a}_1, \cdots, \mathbf{s}_T, \mathbf{a}_T, \mathbf{r}_T}_{\text{last trajectory}}) \text{ st } \mathbf{a}_t \sim \pi_{\text{LLM}}(\cdot | \mathbf{s}_t, \mathbf{c}_n) \ \mathbf{s}_{t+1} \sim \mathbb{P}(\cdot | \mathbf{s}_t, \mathbf{a}_t) \quad (4)$$

The website url is $\mathbf{w}$, the initial task is $\mathbf{c}_0$, the trajectory includes states $\mathbf{s}_t$, actions $\mathbf{a}_t$, judge score $\mathbf{r}_T$, and a harder, grounded task $\mathbf{c}_{n>0}$ for the next loop. Highlighted in Figure 2, we annotate 150k sites with tasks by scaling Equation 4, and release them on huggingface at: *data-for-agents/insta-150k-v2*.

**Model Details.** We utilize pretrained and frozen language models that conform to a chat interface and accept system, user, and assistant prompts. The task proposer system prompt is listed in Appendix 11, and details all cases for which sites are considered unsafe. We employ the Llama 3.1 family of LLMs from Meta (Grattafiori et al., 2024; Touvron et al., 2023b;a), the GPT family of LLMs from OpenAI, and the Gemini family of LLMs from Google. Inference is served using vLLM (Kwon et al., 2023) for the Llama series of models. We employ a sampling temperature of 0.5 and a maximum budget of 1024 new generated tokens, while all other parameters are kept as defaults in the OpenAI chat completions API, which is used to make inference calls to all LLMs.

**Prompt Details.** The task proposer operates in two phases. In an initial phase when just the url of a website is observed, the task proposer generates an initial task $\mathbf{c}_0 \sim \psi_{\text{LLM}}(\mathbf{c}_0 | \mathbf{w})$, and can mark a website as unsafe by setting $\mathbf{c}_0 = \text{N/A}$. The system prompt for this phase is listed in

Figure 3: **Most frequent words in our tasks.** This wordcloud shows the top 500 most frequent words in tasks from the training set of our official huggingface dataset. The size of each word corresponds to its frequency in the dataset. Our tasks span diverse categories and lexicon.

Appendix 11. Agents discussed in Section 5 explore 150k sites annotated with initial tasks $c_0$ and produce trajectories. In a second phase of task generation, we prompt the task proposer with the website url $w$, the initial task $c_0$, the trajectory $s_1, a_1, \cdots, s_T, a_T$, and a system prompt that instructs the LLM to produce a reasoning trace that contains a harder, grounded task $c_{n>0}$. The system prompt for the second phase of task generation is listed in Appendix 11. The refined tasks produced by this iterative process lead to broadly capable agents, demonstrated in Section 6 by our ability to zero-shot transfer agents trained on our data to Web Voyager (He et al., 2024) and compete with frontier LLMs.

The design of the task proposer as a feedback process is important, and the full potential of this design will be realized in future work that trains agents with an on-policy reinforcement learning algorithm. In such future work, the task proposer can be used within the RL loop to generate incrementally harder tasks as agents learn. For this paper, we employ one loop of task generation.

## 4.2 SAFETY & RELIABILITY

Safety is critical when building autonomous agents. The internet contains significant amounts of content that should be removed from training data, in order to avoid agents learning harmful behaviors. To understand the robustness of our safety filter, we input 100 carefully selected websites to the task proposer, of which 50 contain harmful, or mature content. Table 1 reports the accuracy, precision, and recall of the safety filter on this data. For a variety of LLMs, the safety filter displays high accuracy—up to 97% of websites are correctly classified, and recall for detecting unsafe websites is as high as 1.0, suggesting that nearly all unsafe websites are detected by the safety filter.

| Method | Acc. | Prec. | Recall |
|---|---|---|---|
| *Llama 3.1 70B* | 85% | 0.77 | **1.00** |
| *GPT-4o* | 95% | 0.91 | **1.00** |
| *Gemini 1.5 Pro* | **97%** | **0.96** | 0.98 |

| Method | Verifiable Rate |
|---|---|
| *Llama 3.1 70B* | 75% |
| *GPT-4o* | 85% |
| *Gemini 1.5 Pro* | **89%** |

Table 1: **The safety filter displays a high accuracy.** We measure the accuracy, precision, and recall of the safety filter on a set of 100 websites, where 50 contain harmful, or mature content. Up to 97% of websites are correctly classified, and recall is as high as 1.0.

Table 2: **Generated tasks are typically achievable.** We measure the rate that human workers were able to complete and verify their completion of tasks produced by the task proposer for a set of 100 websites. Up to 89% of tasks are achievable, and verifiable.

Reliability is equally important for autonomous agents. Instructions should be followed faithfully by agents, and this requires training them with tasks that are achievable, and verifiable. Table 2 reports the rate that human workers were able to complete and verify their completion of tasks produced by the task proposer in its initial phase. Up to 89% of tasks are achievable, and verifiable according to the study, which suggests the pipeline is producing reliable tasks. Together with results in Section 6, it is likely that data from the InSTA pipeline leads to agents that follow instructions faithfully.

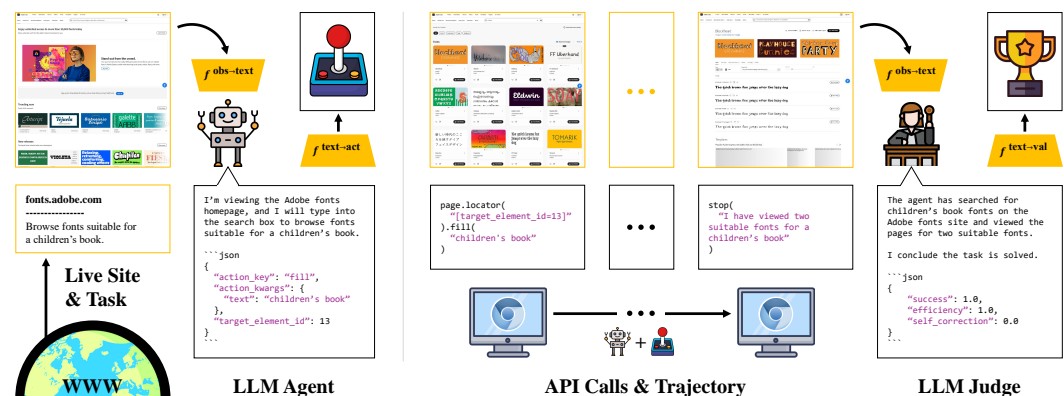

Figure 4: **Automatic evaluation for agents with language model judges.** Building on the large and diverse set of tasks generated by the pipeline, we employ pretrained language models to attempt and evaluate web navigation tasks. We dispatch language model agents to perform tasks by making calls to the Playwright API. We then employ language model judges to evaluate the trajectories.

### 4.3 SCALING TO 150,000 WEBSITES

We leverage Common Crawl for task generation. As of May 2025, the latest web graph released by The Common Crawl Foundation (2025) contains more than 300M unique hosts, which we adapt into a data source for agents. In particular, we select the top 1M sites based on their PageRank values. Common Crawl likely contain many unsafe websites, and these are filtered out by the task proposer. Each phase of task generation requires 14 hours of compute time using two 8-GPU v100 machines, and filters to 150k safe websites annotated with tasks. Statistics of tasks are shown in Figure 3.

## 5 INTERNET-SCALE ENVIRONMENT

By this point, we have reached our first goal—to improve coverage of real-world sites by annotating 150k diverse sites with challenging agentic tasks. To reach our second goal, and move beyond a static dataset, towards a *dynamic internet-scale environment*, we require a robust evaluator for these tasks. Evaluation presents a subtle challenge. The web evolves constantly, and daily changes in website content may invalidate a fixed ground truth reference solution. Driven by necessity, this environment must be evaluated by a model that judges whether an agent's solution is correct, in the latest context.

### 5.1 EVALUATION WITH LANGUAGE MODELS

We model the process of evaluating trajectories from agents as a classification problem, where the goal is to estimate the probability $\mathbf{r}_T$ that a task $\mathbf{c}_n$ is solved, and generate $\mathbf{r}_T$ via next-token prediction, conditioned on a system prompt $\mathbf{x}_{\text{judge}}$, a task $\mathbf{c}_n$, and a trajectory $\mathbf{s}_1, \mathbf{a}_1, \cdots, \mathbf{s}_T, \mathbf{a}_T$. The LLM is instructed to produce a reasoning trace that scores the agent on a scale from 0 to 1, and embed the score as JSON in a fenced code block. We employ a regex template that matches to the first code block in the response, and a JSON decoder to parse $\mathbf{r}_T$ from the response, given by $f^{\text{text} \to \text{val}}(\cdot)$.

$$\mathbf{r}_T = f^{\text{text} \to \text{val}}( \text{ LLM}( \mathbf{x}_{\text{judge}}, \mathbf{c}_n, \text{Enc}(\mathbf{s}_1), \mathbf{a}_1, \cdots, \text{Enc}(\mathbf{s}_T), \mathbf{a}_T ) ) \tag{5}$$

**Verifying The Judge.** To understand the robustness of the judge, we measure its accuracy detecting successful trajectories that were annotated by human workers. We annotate 100 trajectories with binary success labels, and apply a threshold $\mathbf{r}_T > 0.5$ to obtain binary predictions from the judge. Figure 5 reports the accuracy of the judge as a function of the PageRank values of sites, and as a function of the confidence of the judge, given by $\texttt{conf} = 2 \cdot |\mathbf{r}_T - 1/2|$. For all LLMs tested, the judge shows a high accuracy, ranging from $78.0\%$ for *Gemini 1.5 Pro*, to $81.7\%$ for *Llama 3.1 70B*, and $82.6\%$ for *GPT-4o*. Accuracy is stable as PageRank falls, suggesting the judge is accurate for less popular sites that may not be well represented in the LLM's training data. Shockingly, the confidence predicted by LLMs is highly interpretable, and correlates with accuracy to the point that trajectories where $\texttt{conf} = 1$ are classified with an accuracy up to $93.1\%$. The emergent robustness of the judge equips us to efficiently and accurately verify agent solutions on a dynamic internet.

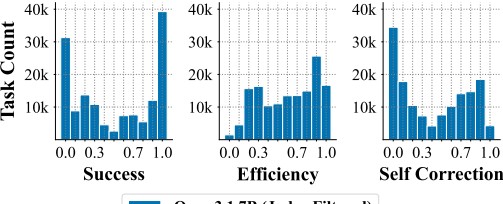

Figure 5: **Language models are robust evaluators.** We measure the accuracy of language models for detecting successful trajectories, and find that accuracy remains stable relative to PageRank values (*left plot*). As models become more confident, their accuracy improves (*right plot*), suggesting confidence is a useful proxy for the reliability of their predictions.

Figure 6: **Statistics for our agent reasoning dataset.** We conduct a large data collection experiment using our top checkpoint for *Qwen3 1.7B*. Our dataset has 2.2M screenshots, 2.2M reasoning traces for actions, and 150k traces for the judge. 50.0% of the trajectories are successful according to the judge, and have diverse ratings for efficiency and self-correction.

## 5.2 SCALING TO 150,000 AGENTS

With our task proposer, agent, and judge driven by pretrained language models, we have all components needed to harness internet-scale data. We conduct a large data collection experiment, running language model agents to complete tasks on 150k websites from our official dataset, producing 2.2M screenshots and 2.2M reasoning traces for actions within 150k trajectories. The judge annotates these trajectories, producing 150k reasoning traces for evaluations, and leading to the statistics in Figure 6. For this experiment, we employ a fine-tuned *Qwen3 1.7B* as the agent (refer to the next section), and *Qwen3 235B* zero-shot as the judge. Data collection requires 1,200 v100 GPU hours, and costs $521.55 based on current AWS spot instance pricing, a fraction of the budget that industry labs are spending towards agents. If you have the right data, this small budget is sufficient and no human annotations are required to build models that compete with frontier LLMs as agents.

## 6 TRAINING AGENTS

We've built an internet-scale data flywheel that produces trajectories annotated with scores from a judge that can help us train LLM agents. To understand the quality of data produced by the flywheel, we conduct a series of experiments training models on the data, and testing on popular benchmarks. These experiments focus on three main questions: *(1) what is the impact of increasing data scale? (2) do agents transfer to new domains? (3) do agents scale with test-time compute?*

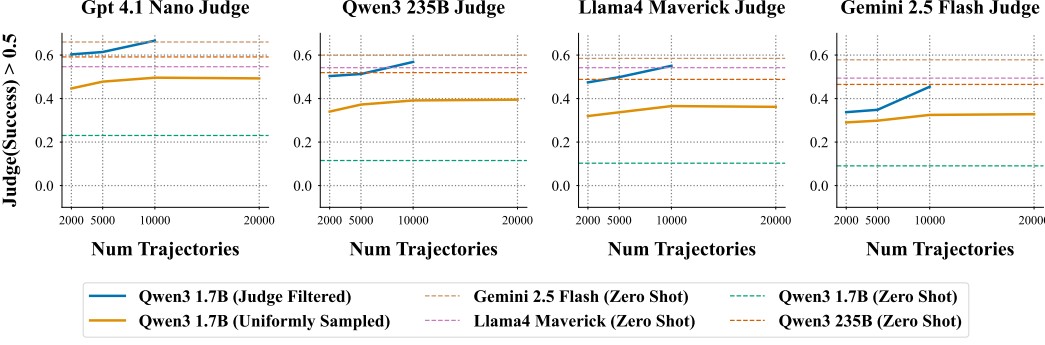

Figure 7: **InSTA unlocks great potential in small models.** We train agents based on *Qwen 3 1.7B* using trajectories produced by a *Qwen 3 235B* data collection policy, and optionally filtered by a *Qwen 3 235B* judge (see Judge Filtered vs. Uniformly Sampled). We report success rates on a test set of 3,000 held-out websites and tasks. Before training, *Qwen 3 1.7B* has a zero-shot success rate of 11.5% according to a *Qwen 3 235B* judge, and we improve this by +45.3% absolute percentage points. Our top checkpoint outperforms the *Qwen 3 235B* data collection policy, and *Llama 4 Maverick*, a frontier LLM with 400B parameters, for which our model is 235 times smaller. Notably, filtering with a *Qwen 3 235B* judge leads to agents that improve according to independent secondary judges, including *Gemini 2.5 Flash*, *Llama 4 Maverick*, and *Gpt 4.1 Nano*, suggesting it generalizes well.

## 6.1 PERFORMANCE IMPROVES WITH DATA SCALE

The recurring lesson in deep learning is that large-scale high-quality data wins, but researchers are struggling to materialize this promise for agents (Xue et al., 2025). Our paper aims to solve the data problem blocking researchers from materializing this promise, and this experiment provides a valuable signal that scaling high-quality data allows small models to compete with strong LLMs from top industry labs. To proceed, we collect 20k trajectories using a *Qwen 3 235B* data collection policy, annotated with scores from a *Qwen 3 235B* judge. We then train agents based on *Qwen 3 1.7B* with SFT on varying scales of the data. Results in Figure 7 show that performance improves with increasing data scale, and gains scale faster on data filtered by the judge. To filter the data, we select trajectories where `Judge(Success) = 1`. Our top checkpoint outperforms the *Qwen 3 235B* data collection policy, and beats *Llama 4 Maverick*, a frontier LLM with 400B parameters, for which our model is 235 times smaller. The trend in the figure suggests there is room to scale further.

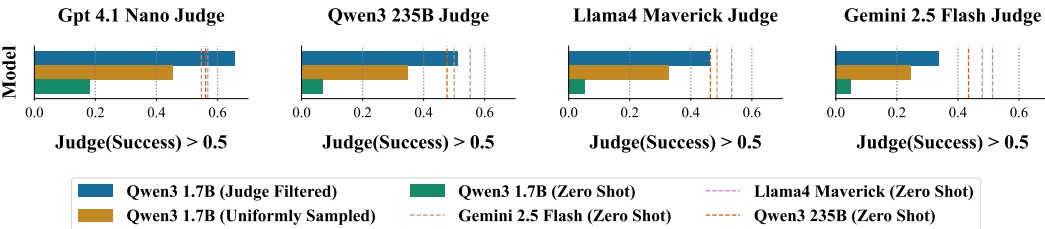

Figure 8: **Our agents zero-shot transfer to WebVoyager.** With no additional training or specialized data, our checkpoints for *Qwen 3 1.7B* in Section 6.1 zero-shot transfer to the WebVoyager benchmark. Trends found on our test set appear to hold for WebVoyager as well, and our top checkpoints for *Qwen 3 1.7B* continue to match frontier LLMs in performance for three of four judges.

## 6.2 AGENTS TRANSFER TO NEW DOMAINS

Statistically correct evaluation for deep learning models requires a test dataset that does not overlap with the training dataset, but researchers do not agree on how to implement this guideline for agents. Recent works train agents on the same websites they test on (Murty et al., 2025; Zhou et al., 2024c; Putta et al., 2024), which may obfuscate progress (Xue et al., 2025). Our next experiment shows that we can implement a stronger train-test split. Agents trained with our data can zero-shot transfer to WebVoyager (He et al., 2024) without any data from the benchmark. Results in Figure 8 show our *Qwen 3 1.7B* checkpoint matching strong LLMs on WebVoyager (He et al., 2024) for three of four judges, confirming trends in Section 6.1. Our ability to zero-shot transfer relatively small agents to WebVoyager (He et al., 2024) makes it likely that our pipeline leads to capable agents.

**Static Benchmarks.** To complement the online evaluation in Figure 8, we also explore how our data impacts agents trained on static benchmarks. We first train baseline agents on human demonstrations from the Mind2Web (Deng et al., 2023) and WebLINX (Lù et al., 2024) datasets. We write a preprocessor to convert our data into the expected action-observation formats these benchmarks use (which involves discarding our reasoning trace). With data converted, we compare the baseline to agents trained on a mix of 80% human data, and 20% our data, and test on (1) their official test set, and (2) 500 diverse sites from our official test set. Results in Figure 9 show that agents trained with our data perform equally well on the original test sets for these benchmark, but generalize better to

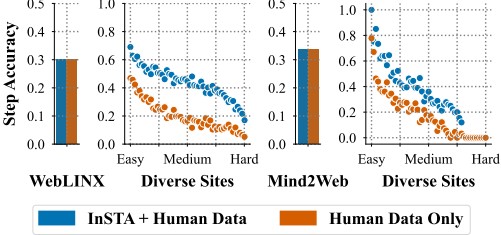

Figure 9: **Our data transfers to static benchmarks.** We train agents with all human data from the WebLINX and Mind2Web training sets, and resulting agents struggle to generalize to more diverse test data. Adding our data improves generalization by +149.0% for WebLINX, and +156.3% for Mind2Web.

our harder test set. Overall, we see +149.0% for WebLINX agents, +156.3% for Mind2Web agents, and gains in *Step Accuracy* on our test set are larger for the harder tasks. Additional experimental details are listed in Appendix H. On three popular benchmarks (WebVoyager, WebLINX, Mind2Web), our pipeline trains capable agents, and does not overly rely on human annotations.

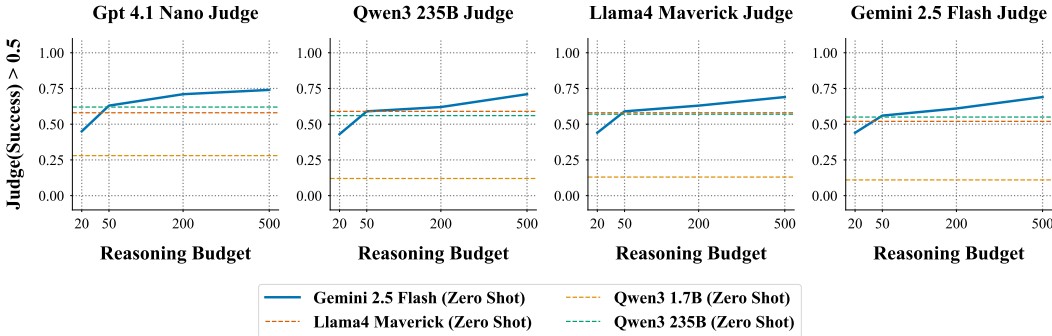

Figure 10: **Agents improve with a larger reasoning budget.** We ablate the number of tokens in the reasoning budget for the top-performing agent, and see a monotonic improvement in the success rate as the reasoning budget increases. *Gemini 2.5 Flash* has a 70% success rate with a budget of 500 reasoning tokens, up from 60% for a budget of 50 tokens. The scaling of performance with the reasoning budget highlights a promising behavior in successful web agents that can be studied.

## 6.3 PERFORMANCE SCALES WITH TEST-TIME COMPUTE

To understand the ability for web agents to scale with additional test-time compute, we ablate the number of tokens in the reasoning budget. Figure 10 shows the success rate as a function of the reasoning budget for *Gemini 2.5 Flash*, the top-performing agent we tested. There is a monotonic improvement in the success rate as the reasoning budget increases, and the trend suggests that performance may not be saturated with 500 reasoning tokens. Training language model agents to reason before taking actions is a promising path to better agents, and we are releasing a large reasoning dataset for multimodal agents to study this. Our dataset contains 2.2M screenshots, 2.2M reasoning traces for actions, 150k traces for judge evaluations, and led to the results in Section 6.1. The data will be linked on our website, alongside an official huggingface dataset for tasks.

## 7 CONCLUSION

In the spirit of deep learning, we have developed an approach to efficiently harness internet data for LLM agents, and have unlocked a *dynamic internet-scale environment*. In building this environment, we presented a method to annotate 150k diverse sites with challenging agentic tasks, and showed how training on data from our pipeline allows small models to compete with frontier LLMs as agents, on a fraction of the budget. Our pipeline consists of a task proposer, agent, and judge driven by pretrained language models that together curate high-quality data for agents, without human intervention.

Our top checkpoint for *Qwen 3 1.7B* has a success rate of 56.9% on our test environment, outperforming the data collection policy *Qwen 3 235B*, beating the 235 times larger *Llama 4 Maverick*, and reaching 94.7% of the performance of *Gemini 2.5 Flash*, while being smaller and faster than these. Our models zero-shot transfer to WebVoyager, and scale with test-time compute. We are releasing the entire pipeline, including code, models and data, so that it may serve as a foundation for researchers to build the next generation of language model agents with internet data.

### 7.1 FUTURE WORK

Our work reveals several exciting directions in future work. First, our work can be scaled further: the latest Common Crawl release contains data for more than 300M sites, suggesting another 1,000 times more data could be available for agents by scaling the pipeline. In addition, we trained agents to optimize the judge scores indirectly via filtered SFT, and the judge's high accuracy suggests that it could be optimized via reinforcement learning instead. RL is especially promising for how it can improve reasoning capabilities in agents. Finally, while the data we collect is multimodal, we focus on textual tasks in this paper, and our pipeline could be extended to produce multimodal tasks.

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

## A  LIMITATIONS & SAFEGUARDS

Language model agents present unique challenges and risks when applied to live tasks on the internet. For instance, agents visiting shopping sites can influence the statistics produced by analytics tools, which can impact prices on products, and product decisions from companies. Furthermore, agents seeing harmful content on the web can add that content to datasets inadvertently, and propagate harmful behaviors to future agents. We mitigate these risks in the design of the task proposal stage. We consider the risks posed to analytics tools by limiting the engagement between agents and sites. We generate only one task per website, and we limit agents to just 30 actions per site, which includes clicks, typing, dropdown selection, and more. By limiting the interaction between agents and sites, the change in website traffic generated by the InSTA pipeline is minimal (just 90 seconds of interaction per site on average). By utilizing data from the InSTA pipeline in an offline fashion, as in Section 6 of the main paper, no additional web traffic is generated when training agents. To ensure that agents do not modify the state of the web (i.e. avoid attempting to make purchases, avoid leaving comments on posts, avoid making accounts, etc), we provide instruct the task proposer (see Figure 2) to avoid writing tasks that require the agent to interact with personal data, or user accounts.

The task proposer is instructed via the system prompt to filter out sites with harmful content, sites not intended for user access, and sites that require making an account to operate, including social media, and forum sites. There is likely a manner to safely train agents to operate user accounts, but we leave this task to future researchers. We explore the performance of the task proposer at filtering out unsuitable sites in Section 4.2, and find that all models detect unsuitable sites with a recall from 0.98 to 1.0, and accuracy up to 97%, suggesting our filter is reliable. Sites used to benchmark the performance of the safety filter are discussed in Appendix E, and thoroughly test the safety filter.

To remove Personally Identifiable Information (PII) from the data used for training agents, we include `scrubadub`, an industry standard PII removal tool for python developed by Leap Beyond, a data consultancy based in the European Union. Our pipeline has an argument that toggles the usage of `scrubadub` to remove PII from all website data, and we recommend this option be set.

## B  ETHICAL CONSIDERATIONS

One important ethical consideration when harnessing internet data is to carefully handle copyrighted, private, and sensitive materials. The internet contains vast amounts of PII, which should be avoided when training models. We address this in two ways. First, we instruct the task proposer to filter out sites that may contain PII, including social media websites, and forums. Second, our pipeline has an argument that toggles the usage of `scrubadub` to remove PII, and we recommend this option be set. To prevent copyrighted materials from being used for training, we do not allow the agent to copy any books, documents, videos, audio, or files. The information present in the HTML DOM is publically available, and is converted into a sparse Markdown summary that focuses on interactive elements, and significantly transforms the underlying webpage in a manner that is not reversible.

These steps significantly reduce, but do not completely eliminate the risk that private, and sensitive materials are shown to the agent, and methods for detecting, replacing, and removing such materials from text and images remains an important task for researchers working on safety.

## C  BROADER IMPACTS

As their capabilities broaden, language models are being used to interface with real-world systems. This shift comes with several benefits and risks. Agents that operate your computer to aid in work tasks can significantly boost productivity for certain workers, but can displace others whose jobs have been fully automated. Agents that operate browsers to complete personal tasks provide convenience, but expose a new attack vector where rogue agents perform unintended actions. Certain risks can be mitigated with proper safeguards, such as post-processing data to prevent jail-breaking, but other risks are existential, harder to address purely with new agent research, and may require policy changes.

```
You are helping us create tasks for a web navigation system. We
    ↪ will tell you the domain of a website. You should provide a
    ↪ realistic, and specific task that a hypothetical user might
    ↪ want to accomplish on that website.

## Skipping Unsafe / Inappropriate Domains

To skip a domain, respond with 'N/A' instead of providing a task.

You should skip domains that have mature, adult, unsafe, or
    ↪ harmful content. If you are unsure whether a domain is safe,
    ↪  please skip it. In addition, skip domains that require
    ↪ logging in or creating an account, such as social media
    ↪ sites, and domains that are not intended for user-access,
    ↪ such as API endpoints and CDNs.

## Here are some domains to provide tasks for:

* 'www.amazon.com': 'Find the price of the 24in LG Ultragear
    ↪ Monitor.'
* 'www.wikipedia.org': 'Look up the history of the Eiffel Tower on
    ↪  Wikipedia.'

## Here are some domains to skip:

* 'fbcdn.net': 'N/A'
* 'api.github.com': 'N/A'

Tasks should not require external knowledge, not modify the state
    ↪ of the web, and should not require logging in or creating an
    ↪  account. For each of the following domains, provide a
    ↪ realistic, and specific task that a user could reasonably
    ↪ accomplish in a single session on the website, and limit
    ↪ your response to 20 words.
```

Figure 11: **System prompt for the exploration phase of task generation**. We design the system prompt for task generation to detect and remove unsafe websites. This prompt ensures that tasks are passive, and do not modify content on a website. Refer to the next figures for the in-context examples used for the task proposer, and the system prompt used in the feedback step.

## D  AGENTS.TXT & STANDARDS FOR INTERNET AGENTS

Akin to `robots.txt` directives, website creators should have a standard format to specify how internet agents are allowed to interact with their websites, and what information on webpages agents are allowed to see. Desireable controls include rate limits for interactions, limits for maximum numbers of interactions, restrictions to allow agents to interact with certain pages and not others, and restrictions on the kind of data on webpages that agents are allowed to observe (achieved via tagging elements to hide their content from agents). In addition to restricting the data available to agents, website creators should have the ability to specify locations for "playgrounds" that replicate certain key functions of their site with virtual tasks and simulated data that are intended to teach agents how to operate their site while directing traffic from agents away from user-facing pages.

## E  MORE DETAILS ON TASK GENERATION

We provide the system prompt used in the first phase of the task generation loop in Figure 11. This prompt was provided to Llama 3.1 70B, GPT-4o, and Gemini 1.5 Pro to generate tasks and filter

sites unsuitable for annotation in Section 4. We carefully designed this system prompt to enforce that generated tasks are passive, and do not modify content on a website. In addition to this system prompt, we employed a list of 100 hand-picked in-context examples of website URLs and appropriate tasks, which are provided in the following JSON list. When querying an LLM, we randomly sample 16 in-context examples from the list, and provide only these examples to the LLM to generate a task to guide exploration of the site. This improves diversity in the exploration phase.

```
[
  {
    "domain": "archive.org",
    "task": "Identify the oldest book available in the public
      ↪ domain on this site."
  },
  {
    "domain": "arxiv.org",
    "task": "Retrieve the latest preprint paper on machine
      ↪ learning."
  },
  {
    "domain": "wikibooks.org",
    "task": "Find a freely available textbook on linear algebra
      ↪ ."
  },
  {
    "domain": "wiktionary.org",
    "task": "Get the definition and etymology of the word '
      ↪ serendipity'."
  },
  {
    "domain": "openlibrary.org",
    "task": "Locate an ebook about classic literature that is
      ↪ available for borrowing."
  },
  {
    "domain": "openculture.com",
    "task": "Find a free online course on ancient history."
  },
  {
    "domain": "theguardian.com",
    "task": "Retrieve an article discussing recent trends in
      ↪ renewable energy."
  },
  {
    "domain": "medium.com",
    "task": "Identify a highly rated blog post on productivity
      ↪ hacks."
  },
  {
    "domain": "goodreads.com",
    "task": "Find the most popular book related to neuroscience
      ↪ ."
  },
  {
    "domain": "wired.com",
    "task": "Retrieve an article about the latest advancements
      ↪ in wearable technology."
  },
  {
```

```
        "domain": "data.gov",
        "task": "Identify the latest government dataset on climate
            ↪ change."
    },
    {
        "domain": "kaggle.com",
        "task": "Find a well-documented data science competition on
            ↪ image recognition."
    },
    {
        "domain": "gov.uk",
        "task": "Locate the latest UK government report on
            ↪ healthcare."
    },
    {
        "domain": "unsplash.com",
        "task": "Find a high-resolution image of the Milky Way
            ↪ Galaxy."
    },
    {
        "domain": "pexels.com",
        "task": "Retrieve a popular photo tagged with 'nature'."
    },
    {
        "domain": "creativecommons.org",
        "task": "Find an article explaining Creative Commons
            ↪ licensing types."
    },
    {
        "domain": "pypi.org",
        "task": "Retrieve the most downloaded Python package for
            ↪ data analysis."
    },
    {
        "domain": "huggingface.co",
        "task": "Identify a popular machine learning model on this
            ↪ platform."
    },
    {
        "domain": "sciencenews.org",
        "task": "Find the most recent article on the health impacts
            ↪ of air pollution."
    },
    {
        "domain": "mit.edu",
        "task": "Retrieve a publicly available research paper on
            ↪ quantum computing."
    },
    {
        "domain": "springer.com",
        "task": "Identify the latest edition of a Springer book on
            ↪ robotics."
    },
    {
        "domain": "jstor.org",
        "task": "Find a research paper discussing the history of the
            ↪  Internet."
    },
    {
```

```
        "domain": "biorxiv.org",
        "task": "Retrieve the most recent bioRxiv preprint on CRISPR
            ↪  technology."
    },
    {
        "domain": "medrxiv.org",
        "task": "Find a public health preprint related to COVID-19."
    },
    {
        "domain": "commons.wikimedia.org",
        "task": "Retrieve a high-resolution image of the Eiffel
            ↪ Tower."
    },
    {
        "domain": "scholar.google.com",
        "task": "Find the most cited article by a specific
            ↪ researcher."
    },
    {
        "domain": "plos.org",
        "task": "Locate the latest research paper on gene editing
            ↪ published here."
    },
    {
        "domain": "flickr.com",
        "task": "Find a photo that has been released under a
            ↪ Creative Commons license."
    },
    {
        "domain": "datacite.org",
        "task": "Retrieve metadata for a dataset related to
            ↪ environmental studies."
    },
    {
        "domain": "orcid.org",
        "task": "Find the ORCID ID of a well-known researcher in AI
            ↪ ."
    },
    {
        "domain": "zotero.org",
        "task": "Retrieve an article discussing citation management
            ↪ tools."
    },
    {
        "domain": "github.com",
        "task": "Find the most starred repository on deep learning."
    },
    {
        "domain": "figshare.com",
        "task": "Retrieve an open dataset on climate patterns."
    },
    {
        "domain": "zenodo.org",
        "task": "Find the latest publication on open science
            ↪ practices."
    },
    {
        "domain": "worldcat.org",
        "task": "Locate a catalog entry for a rare book on botany."
```

```
        },
        {
            "domain": "biodiversitylibrary.org",
            "task": "Retrieve a scanned copy of an 18th-century
                ↪ botanical illustration."
        },
        {
            "domain": "genome.gov",
            "task": "Find the latest update on the Human Genome Project
                ↪ ."
        },
        {
            "domain": "merriam-webster.com",
            "task": "Retrieve the definition and usage of the word '
                ↪ quantum'."
        },
        {
            "domain": "stanford.edu",
            "task": "Find the most recent online lecture on artificial
                ↪ intelligence."
        },
        {
            "domain": "edx.org",
            "task": "Retrieve a TED Talk on leadership in technology."
        },
        {
            "domain": "ted.com",
            "task": "Find the latest ocean temperature data available."
        },
        {
            "domain": "noaa.gov",
            "task": "Retrieve a dataset related to consumer behavior."
        },
        {
            "domain": "data.world",
            "task": "Find a course on data visualization."
        },
        {
            "domain": "curious.com",
            "task": "Retrieve a well-cited article on the psychological
                ↪ impact of social media."
        },
        {
            "domain": "theconversation.com",
            "task": "Identify a recent research paper on biodiversity
                ↪ conservation."
        },
        {
            "domain": "nature.com",
            "task": "Retrieve the latest article on genomics research."
        },
        {
            "domain": "pnas.org",
            "task": "Find a science news article on robotics
                ↪ advancements."
        },
        {
            "domain": "sciencedaily.com",
            "task": "Identify the top story on global health issues."
```

```
    },
    {
        "domain": "bbc.com",
        "task": "Retrieve a recent podcast episode about space
            ↪ exploration."
    },
    {
        "domain": "npr.org",
        "task": "Locate the most recent update on the global
            ↪ biodiversity status."
    }
]
```

We also provide the system prompt used in the second phase of the task generation loop, where trajectories from agents are fed back to the task proposer, which generates a harder, grounded task. This prompt instructs the task proposer to create a challenging task based on how an expert user could be expected to use the shown website. The task proposer also predicts a list on intermediate steps that can be used as a hint for agents, and a success criteria that can be used to improve the verifier.

```
You are a helpful assistant designing tasks for a web automation
    ↪ script. I will show you previous runs of the script,
    ↪ including previous tasks, webpages, actions, and performance
    ↪  reviews, formatted in markdown. Help me design *challenging
    ↪ * new tasks.

## Formatting The Proposed Task

Format your task in the following JSON schema:

```json
{
    "proposed_task": str,
    "steps": List[str],
    "criteria": str
}
```

Here is what each key means:

- `proposed_task`: A specific, challenging task that an expert
    ↪ user might leverage this website to complete.
    - Must not require making an account, logging in, submitting
        ↪ personal information, making a purchase, or placing an
        ↪ order.

- `steps`: Steps an expert user would follow to complete the
    ↪ proposed task.
- `criteria`: The required answer, and criteria to determine if
    ↪ the task was completed.

## Example Tasks For Inspiration

Suppose you want to design a task around the 'C-to-C Hose-Shut-Off
    ↪  Valve' on 'awg-fittings.com':

```json
{
```

```
    "proposed_task": "What is the C-to-C Hose-Shut-Off Valve length
        ↪ in mm?",
    "steps": [
        "Navigate to 'awg-fittings.com'",
        "Open the product catelog for fittings",
        "Locate the product listing for the C-to-C Hose-Shut-Off
            ↪ Valve",
        "Find the product length in mm, and respond with that length
            ↪ in the answer"
    ],
    "criteria": "The answer should include the specific length of
        ↪ '237 mm' for this product"
}
```

Suppose you want to design a task around the document 'The Angora
    ↪ cat; how to breed train and keep it' on 'biodiversitylibrary.
    ↪ org':

```json
{
    "proposed_task": "Open a scanned copy of 'The Angora cat; how
        ↪ to breed train and keep it'.",
    "steps": [
        "Navigate to 'biodiversitylibrary.org'",
        "Search for 'The Angora cat; how to breed train and keep it'
            ↪ in the search bar",
        "Click on the title of the document in the search results",
        "Confirm the correct document is displayed in an embedded
            ↪ PDF reader"
    ],
    "criteria": "The final webpage should display the correct
        ↪ document in an embedded PDF reader"
}
```

Suppose you want to design a task around the 'Generative
    ↪ Adversarial Networks' paper on 'scholar.google.com':

```json
{
    "proposed_task": "How many citations does the paper 'Generative
        ↪ Adversarial Networks' have?",
    "steps": [
        "Navigate to 'scholar.google.com'",
        "Search for 'Generative Adversarial Networks' in the search
            ↪ bar",
        "Locate the correct paper in the search results",
        "Find an up-to-date citation count, and respond with that
            ↪ count in the answer"
    ],
    "criteria": "The answer should include an up-to-date citation
        ↪ count, which is '80613' as of April 2025"
}
```

Suppose you want to design a task around the word 'serendipity' on
    ↪ 'wiktionary.org':
```

```json
{
    "proposed_task": "What is the definition and etymology of the
        ↪ word 'serendipity'?",
    "steps": [
        "Navigate to 'wiktionary.org'",
        "Search for 'serendipity' in the search bar",
        "Find the definition and etymology sections of the '
            ↪ serendipity' page",
        "Summarize the contents of these sections in the answer"
    ],
    "criteria": "The answer should mention Serendip (or Serendib),
        ↪ coined by English writer and politician Horace Walpole in
        ↪  1754"
}
```

Thanks for helping me design challenging new tasks, please follow
    ↪ the instructions carefully. Start your response with an
    ↪ analysis for how an expert user would leverage this website,
    ↪  followed by a step-by-step breakdown of your proposed task,
    ↪  and finally, enter your task in the JSON format. Respond in
    ↪  500 words.

### E.1 DETAILS FOR SAFETY EXPERIMENT

Using these prompts for task generation, we remove unsafe websites. To evaluate the performance of our filter, we employed a set of 100 curated websites, where 50 are manually verified as safe, and 50 are manually verified as unsafe based on the filtering conditions. These sites were chosen to span popular sites that typical users are likely familiar with, and less popular websites that may be underrepresented in LLM training data.

```
safe_sites_list = ['dhss.mo.gov', 'dizionari.corriere.it', '
    ↪ southgippsland.vic.gov.au', 'ds.iris.edu', 'lobbycontrol.de
    ↪ ', '4rsmokehouse.com', 'barnsleyfc.co.uk', 'wiwi.uni-
    ↪ wuerzburg.de', 'uplandca.gov', 'lsus.edu', 'wpcode.com', '
    ↪ webopedia.internet.com', 'tamko.com', 'premierchristian.news
    ↪ ', 'genome.jgi.doe.gov', 'burgerking.ca', 'thehugoawards.org
    ↪ ', 'radio.fm', 'thevinyldistrict.com', 'unilang.org', '
    ↪ raywhitegroup.com', 'grapevinetexas.gov', 'sanfrancisco.
    ↪ cbslocal.com', 'hyde-design.co.uk', 'breastcancerfoundation.
    ↪ org.nz', 'ludwigsburg.de', 'ignitionrobotics.org', '
    ↪ deliverit.com.au', 'kodokan.org', 'clickstay.com', '
    ↪ searchdatamanagement.techtarget.com', 'oceanario.pt', '
    ↪ wentworthpuzzles.com', 'catholicworldreport.com', 'quizlet.
    ↪ com', 'innovation.nhs.uk', 'synonyms.reverso.net', 'news.
    ↪ siemens.co.uk', 'readability-score.com', 'co.modoc.ca.us', '
    ↪ cityofmyrtlebeach.com', 'loire.gouv.fr', 'lawphil.net', '
    ↪ saem.org', 'parmigianoreggiano.it', 'engaging-data.com', '
    ↪ itf-tkd.org', 'aka.education.gov.uk', 'ub.uni-kl.de', '
    ↪ mottchildren.org']

unsafe_sites_list = ['refer to the code release']
```

We provide the list of unsafe sites in the code to keep this manuscript clean. Using these lists, we conduct an experiment measuring the accuracy, precision, and recall of our safety filter for detecting sites that are not suitable for training agents. In this experiment, we run the initial exploration phase

of the task proposer with the same prompts used in the main experiments (shown in the last section), and we consider a website to be marked positive for unsafe content if the task proposer generates "N/A" rather than a task. We then calculate accuracy, precision, and recall for various LLMs.

## E.2 DETAILS FOR RELIABILITY EXPERIMENTS

We evaluated the verifiable rate of tasks generated by the initial phase of task generation manually. For a set of 100 randomly sampled websites marked as safe by the task proposer in the initial phase, we attempted to complete the generated task, and checked that enough information is present on the website that a solution can be verified. In total, we annotated 300 tasks for Table 2 in 6 hours, and provide the 100 website URLs used in this experiment in the following code block.

```
reliability_sites_list = ['godaddy.com', 'chrome.google.com', '
    ↪ apple.com', 'support.cloudflare.com', 'support.apple.com', '
    ↪ edition.cnn.com', 'go.microsoft.com', 'google.de', 'w3.org',
    ↪  'yandex.ru', 'bfdi.bund.de', 'microsoft.com', 'apps.apple.
    ↪ com', 'networksolutions.com', 'support.mozilla.org', 'yelp.
    ↪ com', 'cnn.com', 'ec.europa.eu', 'developer.mozilla.org', '
    ↪ icann.org', 'books.google.com', 'globenewswire.com', '
    ↪ onlinelibrary.wiley.com', 'gnu.org', 'slideshare.net', '
    ↪ metacpan.org', 'porkbun.com', 'oag.ca.gov', 'spiegel.de', '
    ↪ linuxfoundation.org', 'help.opera.com', 'mayoclinic.org', '
    ↪ podcasts.apple.com', 'nhs.uk', 'addons.mozilla.org', 'google.
    ↪ fr', 'pewresearch.org', 'finance.yahoo.com', 'weforum.org',
    ↪ 'g2.com', 'savethechildren.org', 'news.com.au', 'biblia.com
    ↪ ', 'yr.no', 'engadget.com', 'microsoftstore.com', 'ema.
    ↪ europa.eu', 'theintercept.com', 'princeton.edu', '
    ↪ foodandwine.com', 'sfgate.com', 'voguebusiness.com', '
    ↪ ourworldindata.org', 'livingwage.org.uk', 'cms.law', '
    ↪ msdmanuals.com', 'websitesetup.org', 'support.xbox.com', '
    ↪ treehugger.com', 'tripadvisor.com.pe', 'mondragon.edu', '
    ↪ greenparty.ca', 'aaojournal.org', 'restaurantpassion.com', '
    ↪ iwillteachyoutoberich.com', 'moneyconvert.net', '
    ↪ gesundheitsinformation.de', 'ovc.uoguelph.ca', 'zdnet.be', '
    ↪ oxfordamerican.org', 'snackandbakery.com', 'journals.uic.edu
    ↪ ', 'confused.com', 'standards.globalspec.com', '
    ↪ onlyinyourstate.com', 'ahsgardening.org', 'wyze.com', '
    ↪ nornickel.ru', 'viessmann.fr', 'benetton.com', 'firecomm.gov.
    ↪ mb.ca', 'executedtoday.com', 'eukn.eu', 'fraeylemaborg.nl',
    ↪ 'verizon.com/about/news-center', 'orthodoxalbania.org', '
    ↪ cheapjoes.com', 'bake-eat-repeat.com', '
    ↪ plattformpatientensicherheit.at', 'hifinews.com', '
    ↪ cellsignal.com', 'thenotariessociety.org.uk', 'chosenfoods.
    ↪ com', 'westerndressageassociation.org', 'pridesource.com', '
    ↪ northtacomapediatricdental.com', 'strade-bianche.it', '
    ↪ pvdairport.com', 'institute.sandiegozoo.org', 'raintaxi.com
    ↪ ']
```

## E.3 AUTOMATIC TASK CATEGORIZATION

We employ *Llama 3.1 70B* to categorize tasks. We prompt *Llama 3.1 70B* with the system prompt in Figure 12 to assign a category in 3 words or less to encourage simple categories. Categories have 16.9 tasks on average, and 953 categories have more than the mean, while 7741 have less than the mean. There is occasional overlap between categories, which can be observed in Figure 13, but for the purposes of understanding performance by category, overlap is acceptable provided categories have sufficiently large numbers of tasks, and performance per category can be accurately calculated. We provide our task categorization script in the official code release.

```
You are a helpful scientific assistant categorizing tasks on the
    ↪ web. You will observe a domain and web navigation task, and
    ↪ you should provide a concise categorization of the task in 3
    ↪  words or less. For example, if the domain is "google.com"
    ↪ and the task is "find a recipe for mashed potato", you may
    ↪ categorize the task as "recipe search".

## Task Format

Here is the format for the task:

[domain]: [task]

Here is what each part means:

`[domain]`: The domain of the website you are observing.
`[task]`: The task a user is trying to accomplish on the website.

## Response Format

Respond with a category name for the task in 3 words or less, and
    ↪ provide only the category name, do not provide an
    ↪ explanation or justification for the categorization.

Here is the next task, please follow the instructions carefully.
```

Figure 12: **System prompt for task categorization**. We employ *Llama 3.1 70B* to automatically label task categories for our dataset. We prompt the LLM to assign categories in 3 words or less, and set the sampling temperature to 0.5 to encourage predictions to use more consistent language. Using these categories, we seek to understand agent performance by category.

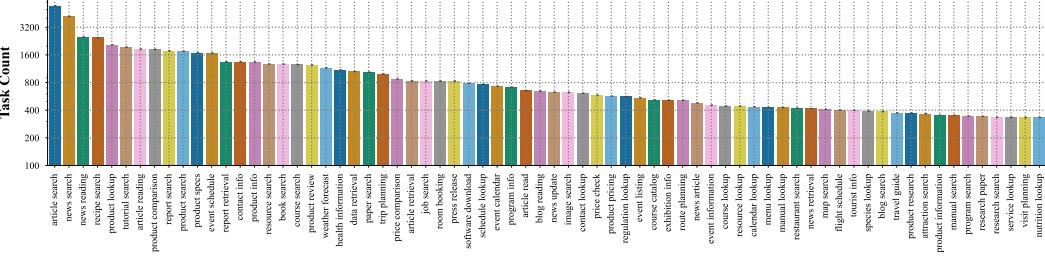

Figure 13: **Largest categories for task generation**. We categorize 150k tasks generated by our pipeline in Section 4, and visualize the number of tasks in the largest 70 categories. Top categories include *article search*, *news search*, *recipe search*, and *product lookup*. The top 12 task categories have more than 1600 tasks assigned to each of them, the mean number of tasks per category is 16.9, and 89% of categories (7741 in total) have fewer than the mean number of tasks.

# F  UNDERSTANDING AGENT CAPABILITIES & LIMITATIONS

To complement the analyses presented in Section 5, we explore the categories of tasks that agents succeed at most frequently. Shown in Figure 14, we plot the average judge success probability prediction $\mathbf{r}_T$ versus task category for the top 70 most successful categories that have at least 100 tasks assigned to them. Based on the figure, top categories include searching for *contact information*, finding *hours of operation*, looking up *biographical information*, obtaining current *weather forecasts*, and conducting *health research*. Based on these results, the top 22 categories are solved with an average success probability $> 0.5$ using zero-shot agents based on *Llama 3.1 70B*. As stronger

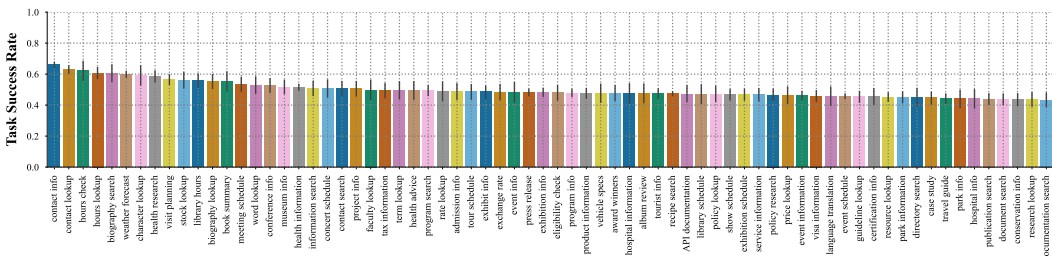

Figure 14: **Most solved categories for task generation**. We explore the completion rates for the top categories of tasks generated by our pipeline. We restrict our focus to categories where at least 100 tasks are assigned, and plots the success rates for the top 70 categories. Results show that 22 of these categories are solved with more than a 50% rate with zero-shot agents based on *Llama 3.1 70B*.

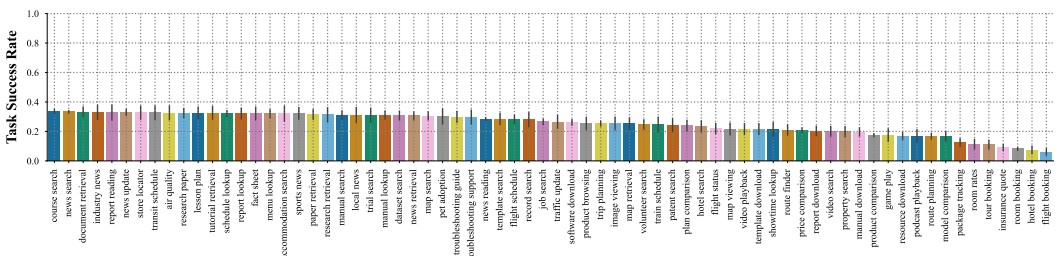

Figure 15: **Least successful categories for internet-scale task generation**. Similar to the previous figure, we explore the rates of task completion for the bottom 70 categories that have at least 100 tasks assigned to them. While the majority of the least successful categories have success rates greater than 20%, performance drops as low as 5%. Many of the categories shown in the plot above involve actions that are not feasible given the current limitations of the Playwright API, and may be possible in future work that extends agents to a fully-operable virtual computer environment. In addition, better LLM backbones are likely to improve performance.

models are developed, the success rates for agents running in our pipeline are likely to improve, and the quality of the data we generate will jointly improve.

In addition to studying the best-performing categories, we also explore the limitations of current agents via their least successful categories. Shown in Figure 15, we select the bottom 70 categories via their average judge success probability for categories with at least 100 tasks assigned. Many of these categories require agents to remember and reason about previous interactions, such as the *product comparison* category. For this category, an agent must review several products, and compare details from memory. In these cases, access to a note-taking tool may improve performance. Additionally, certain task categories involve requests that are not feasible given the limitations of the Playwright API, including categories for *downloading reports / manuals*, and *opening files*. While these tasks are not currently feasible, providing agents with a fully-operable virtual computer environment with applications pre-installed could unlock these abilities in future work.

## G  AGENT, JUDGE & TASK PROPOSER SYSTEM PROMPTS

We provide the system prompt that powers the agent in this paper. This prompt is released in the code, alongside a fast HTML to Markdown processor we built. The agent prompt is carefully designed to elicit reasoning capabilities, and the experiment in Figure 10 shows the prompt is successful.

```
You are helping me complete tasks by operating a web browser. I
    ↪ will share the current task, and a sequence of webpages and
    ↪ actions from previous steps.

## Action Instructions
```

Based on the information we discovered so far, and the progress we
    ↪  made in previous steps, you are helping me determine the
    ↪ next action.

You will provide an action as JSON in a fenced code block:

```json
{
    "action_key": str,
    "action_kwargs": dict,
    "target_element_id": int | null
}
```

Actions have the following components:

- `action_key`: The name of the selected action.
- `action_kwargs`: A dictionary of arguments for the action.
- `target_element_id`: An optional id for the element to call the
    ↪ action on.

## Action Definitions

I've prepared an API documentation below that defines the actions
    ↪ we can use to complete the task.

### Click Action Definition

- `click`: Click on an element specified by `target_element_id`.

### Example Click Action

Suppose you want to click `[id: 5] Sales link`:

```json
{
    "action_key": "click",
    "action_kwargs": {},
    "target_element_id": 5
}
```

### Hover Action Definition

- `hover`: Hover over an element specified by `target_element_id`

### Example Hover Action

Suppose you want to hover over `[id: 2] Company Logo image`:

```json
{
    "action_key": "hover",
    "action_kwargs": {},
    "target_element_id": 2
}
```

```
### Scroll Action Definition

- `scroll`: Scroll the page by `delta_x` pixels to the right and `
  ↪ delta_y` pixels down.
    - `delta_x`: The number of pixels to scroll to the right.
    - `delta_y`: The number of pixels to scroll down.

### Example Scroll Action

Suppose you want to scroll down the page by 300 pixels:

```json
{
    "action_key": "scroll",
    "action_kwargs": {
        "delta_x": 0,
        "delta_y": 300
    },
    "target_element_id": null
}
```

### Fill Action Definition

- `fill`: Fill an input element specified by `target_element_id`
  ↪ with text.
    - `value`: The text value to fill into the element.

### Example Fill Action (Text Input)

Suppose you want to fill `[id: 13] "Name..." (Enter your name text
  ↪  input)` with the text `John Doe`:

```json
{
    "action_key": "fill",
    "action_kwargs": {
        "value": "John Doe"
    },
    "target_element_id": 13
}
```

### Example Fill Action (Range Slider)

Suppose you want to set `[id: 71] "$250 (5)" (range slider min: 0
  ↪ max: 50 step: 1)` to the value of `$1000`. The slider has a
  ↪ range of 0 to 50 with a step of 1, and the value is
  ↪ currently set to `5`. You must translate the desired `$1000`
  ↪  to the correct underlying value of `20`:

```json
{
    "action_key": "fill",
    "action_kwargs": {
        "value": "20"
    },
    "target_element_id": 71
}
```

```
```

### Select Action Definition

- `select`: Select from a dropdown element specified by `
  ↪ target_element_id`.
    - `label`: The option name to select in the element.

### Example Select Action

Suppose you want to select the option `red` from `[id: 67] "blue"
    ↪ (color select from: red, blue, green)`:

```json
{
    "action_key": "select_option",
    "action_kwargs": {
        "label": "red"
    },
    "target_element_id": 67
}
```

### Set Checked Action Definition

- `set_checked`: Check or uncheck a checkbox specified by `
  ↪ target_element_id`.
    - `checked`: Boolean value to check or uncheck the checkbox.

### Example Set Checked Action

Suppose you want to check `[id: 21] "I agree to the terms and
    ↪ conditions" (checkbox)`:

```json
{
    "action_key": "set_checked",
    "action_kwargs": {
        "checked": true
    },
    "target_element_id": 21
}
```

### Go Back Action Definition

- `go_back`: Go back to the previous page (`target_element_id`
  ↪ must be null).

### Example Go Back Action

Suppose you want to go back to the previous page:

```json
{
    "action_key": "go_back",
    "action_kwargs": {},
    "target_element_id": null
}
```

```
```

### Goto Action Definition

- `goto`: Navigate to a new page (`target_element_id` must be null
  ↪ ).
    - `url`: The URL of the page to navigate to.

### Example Goto Action

Suppose you want to open the DuckDuckGo search engine:

```json
{
    "action_key": "goto",
    "action_kwargs": {
        "url": "https://www.duckduckgo.com"
    },
    "target_element_id": null
}
```

### Stop Action Definition

- `stop`: Stop when the task is complete, and report your progress
  ↪ .
    - `answer`: Optional answer sent back to me.

### Example Stop Action

Suppose the task is complete, and you want to stop and report your
  ↪ progress:

```json
{
    "action_key": "stop",
    "action_kwargs": {
        "answer": "The desired task is now complete."
    },
    "target_element_id": null
}
```

## Formatting Your Response

Write a 200 word revised plan based on new information we
  ↪ discovered, and progress we made in previous steps. After
  ↪ your response, provide the next action as JSON in a fenced
  ↪ code block.
```

We also provide the system prompt used by the judge. The system prompt instructs the judge to predict JSON within a fenced code block that contains a "success" key, an "efficiency" key, and a "self_correction" key. The success key represents a score from 0 to 1 that estimates the probability the task is successfully completed. The efficiency key represents a score from 0 to 1 that estimates the probability the agent has taken the most efficient path to solve the task. The self correction key represents a score from 0 to 1 that estimates the probability that the agent has demonstrated self corrective behaviors during its completion of the task. These behaviors include when the agent backtracks to a more promising state, re-plans when new information is discovered relevant to the task, and recognizes its own mistakes. These are generally behaviors we expect from successful

agents, but for this paper we only filter by the success key to select training data for agents.

```
You are helping me evaluate a browser automation script. I will
    ↪ share a task provided to the script, and a sequence of
    ↪ webpages and actions produced by the script.

## The Action Format

The script produces actions as JSON in a fenced code block:

```json
{
    "action_key": str,
    "action_kwargs": dict,
    "target_element_id": int
}
```

Actions have the following components:

- `action_key`: The name of the selected action.
- `action_kwargs`: A dictionary of arguments for the action.
- `target_element_id`: An optional id for the element to call the
    ↪ action on.

## Action Definitions

I've prepared an API documentation below that defines the actions
    ↪ the script can use to complete the task.

### Click Action Definition

- `click`: Click on an element specified by `target_element_id`.

### Example Click Action

Here is an example where the script clicks `[id: 5] Sales link`:

```json
{
    "action_key": "click",
    "action_kwargs": {},
    "target_element_id": 5
}
```

### Hover Action Definition

- `hover`: Hover over an element specified by `target_element_id`

### Example Hover Action

Here is an example where the script hovers over `[id: 2] Company
    ↪ Logo image`:

```json
{
    "action_key": "hover",
```

```
    "action_kwargs": {},
    "target_element_id": 2
}
```

### Scroll Action Definition

- `scroll`: Scroll the page by `delta_x` pixels to the right and `
  ↪ delta_y` pixels down.
  - `delta_x`: The number of pixels to scroll to the right.
  - `delta_y`: The number of pixels to scroll down.

### Example Scroll Action

Here is an example where the script scrolls down the page by 300
  ↪ pixels:

```json
{
    "action_key": "scroll",
    "action_kwargs": {
        "delta_x": 0,
        "delta_y": 300
    },
    "target_element_id": null
}
```

### Fill Action Definition

- `fill`: Fill an input element specified by `target_element_id`
  ↪ with text.
  - `value`: The text value to fill into the element.

### Example Fill Action (Text Input)

Here is an example where the script fills `[id: 13] "Name..." (
  ↪ Enter your name text input)` with the text `John Doe`:

```json
{
    "action_key": "fill",
    "action_kwargs": {
        "value": "John Doe"
    },
    "target_element_id": 13
}
```

### Example Fill Action (Range Slider)

Here is an example where the script sets `[id: 71] "$250 (5)" (
  ↪ range slider min: 0 max: 50 step: 1)` to the value of `$1000
  ↪ `. This slider has a range of 0 to 50 with a step of 1, and
  ↪ the value is currently set to `5`. The script translates the
  ↪  desired `$1000` to the correct underlying value of `20`:

```json
{
```

```
1674      "action_key": "fill",
1675      "action_kwargs": {
1676          "value": "20"
1677      },
1678      "target_element_id": 71
1679  }
1680  ```
1681
1682  ### Select Action Definition
1683
1684  - `select`: Select from a dropdown element specified by `
1685    ↪ target_element_id`.
1686      - `label`: The option name to select in the element.
1687  ### Example Select Action
1688
1689  Here is an example where the script selects the option `red` from
1690    ↪ `[id: 67] "blue" (color select from: red, blue, green)`:
1691
1692  ```json
1693  {
1694      "action_key": "select_option",
1695      "action_kwargs": {
1696          "label": "red"
1697      },
1698      "target_element_id": 67
1699  }
1700  ```
1701  ### Set Checked Action Definition
1702
1703  - `set_checked`: Check or uncheck a checkbox specified by `
1704    ↪ target_element_id`.
1705      - `checked`: Boolean value to check or uncheck the checkbox.
1706  ### Example Set Checked Action
1707
1708  Here is an example where the script checks `[id: 21] "I agree to
1709    ↪ the terms and conditions" (checkbox)`:
1710
1711  ```json
1712  {
1713      "action_key": "set_checked",
1714      "action_kwargs": {
1715          "checked": true
1716      },
1717      "target_element_id": 21
1718  }
1719  ```
1720
1721  ### Go Back Action Definition
1722
1723  - `go_back`: Go back to the previous page (`target_element_id`
1724    ↪ must be null).
1725
1726  ### Example Go Back Action
1727
1728  Here is an example where the script goes back to the previous page
1729    ↪ :
```

```json
{
    "action_key": "go_back",
    "action_kwargs": {},
    "target_element_id": null
}
```

### Goto Action Definition

- `goto`: Navigate to a new page (`target_element_id` must be null
  ↪ ).
    - `url`: The URL of the page to navigate to.

### Example Goto Action

Here is an example where the script opens DuckDuckGo search:

```json
{
    "action_key": "goto",
    "action_kwargs": {
        "url": "https://www.duckduckgo.com"
    },
    "target_element_id": null
}
```

### Stop Action Definition

- `stop`: Stop when the task is complete, and report the progress.
    - `answer`: Optional answer from the script.

### Example Stop Action

Here is an example where the script stops and reports its progress
  ↪ :

```json
{
    "action_key": "stop",
    "action_kwargs": {
        "answer": "The desired task is now complete."
    },
    "target_element_id": null
}
```

## Evaluation Instructions

Based on the progress of the script, you are helping me determine
  ↪ if the desired task has been completed successfully.

You will provide scores as JSON in a fenced code block:

```json
{
    "success": float,
```

```
    "efficiency": float,
    "self_correction": float
}
```

### Score Definitions

- `success`: Your confidence the desired task has been completed
  ↪ successfully.
    - range: 0.0 (not possible) to 1.0 (absolutely certain).

- `efficiency`: Your confidence the script has taken the most
  ↪ efficient path to complete the task.
    - range: 0.0 (not possible) to 1.0 (absolutely certain).

- `self_correction`: Your confidence the script has demonstrated
  ↪ self-corrective behaviors during its completion of the task.
  ↪  These behaviors include backtracking to a more promising
  ↪ state, replanning when new information is discovered, and
  ↪ recognizing its own mistakes.
    - range: 0.0 (not possible) to 1.0 (absolutely certain).

Write a 300 word analysis that establishes specific criteria to
  ↪ rigorously evaluate whether the task was completed, followed
  ↪  by which criteria the script has satisfied. After your
  ↪ response, provide your scores as JSON in a fenced code block.
  ↪
```

Finally, we provide the system prompt used in the task proposer to refine the task generated by the first iteration, and to raise the difficulty. The task proposer is instructed via the system prompt to produce JSON with a "proposed_task" key that represents the task for the agent to complete, a "steps" key that represents the steps that an agent would follow to complete the task, and a "criteria" key that represents the criteria tha judge will employ to determine if the task has been completed.

```
You are helping me instruct a language model agent that interacts
  ↪ with and navigates live webpages. We instructed the agent to
  ↪  complete an initial task, and I will share a sequence of
  ↪ webpages visited by the agent during its operation.

## Your Instructions

Help me refine the task, steps and criteria to raise the
  ↪ difficulty, while balancing the agent's capacity to
  ↪ successfully complete the task.

You will provide a task as JSON in a fenced code block:

```json
{
    "proposed_task": str,
    "steps": List[str],
    "criteria": List[str]
}
```

Tasks have the following components:

- `proposed_task`: Instruct the agent to complete a task for you
  ↪ as if you are a real user that wants help on the website.
```

```
- 'steps': Precise steps in an efficient trajectory that completes
    ↪ the task.
- 'criteria': Ground truth answers and criteria to determine if
    ↪ the agent completes the task.

Tasks must adhere to the following guidelines:

- Must not require logging in, or making an account.
- Must not require making a purchase, booking, or placing an order
    ↪ .
- Must not require creating, deleting, or modifying any posts,
    ↪ articles, or webpages.

## Example Tasks

I've prepared some examples to inspire your task design.

### 'liveevents.iadb.org'

In this example, we explored 'liveevents.iadb.org' and saw an
    ↪ event page for the IDB Annual Meetings, which includes a
    ↪ list of the official hotels and instructions for official
    ↪ delegations.

```json
{
    "proposed_task": "I'm attending the IDB Annual Meetings and
        ↪ need to find accommodation. Please provide the address
        ↪ and phone number for the 'Pullman Santiago Vitacura' and
        ↪ 'Double Tree by Hilton' hotels. Additionally, what
        ↪ specific details do official delegations need to provide
        ↪ to access their special hotel block?",
    "steps": [
        "Navigate to 'https://liveevents.iadb.org'.",
        "Click on the 'Hotels' link in the navigation menu.",
        "Locate 'Pullman Santiago Vitacura' in the 'OFFICIAL HOTELS
            ↪ FOR THE ANNUAL MEETINGS' list and extract its address
            ↪ and telephone number.",
        "Locate 'Double Tree by Hilton' in the same list and extract
            ↪  its address and telephone number.",
        "Read the instructions under 'HOTELS FOR OFFICIAL
            ↪ DELEGATIONS' to identify the required information for
            ↪ accessing the special hotel block.",
        "State the addresses and telephone numbers for both hotels
            ↪ and the required information for official delegations
            ↪ ."
    ],
    "criteria": [
        "The agent successfully navigates to the 'Hotels' page on '
            ↪ liveevents.iadb.org'.",
        "The address for Pullman Santiago Vitacura is stated as '
            ↪ Avenida Vitacura 3201 Vitacura, 7630578 Santiago,
            ↪ Chile'.",
        "The telephone number for Pullman Santiago Vitacura is
            ↪ stated as '+56 2 2944 7800'.",
        "The address for Double Tree by Hilton is stated as 'Avenida
            ↪  Vitacura 2727, Las Condes Santiago, Chile'.",
        "The telephone number for Double Tree by Hilton is stated as
            ↪  '+56 2 2587 7000'.",
```

```
        "The agent states that official delegations need to include
            ↪ 'the name of your country' and 'the code (included in
            ↪ the invitation letters to the Governors)' to access
            ↪ the special hotel block."
    ]
}
```

### `boldtcastle.com`

In this example, we explored `boldtcastle.com` and saw a page with
    ↪  information about visiting Boldt Castle, including
    ↪ operating dates, admission prices, and how to get to Heart
    ↪ Island.

```json
{
    "proposed_task": "Help me plan a visit to Boldt Castle in 2025
        ↪ with one adult and one 6-year-old. Please provide the
        ↪ operating dates and hours for the 2025 season, the
        ↪ admission cost for just the castle for both of us, how to
        ↪  get to Heart Island, and the best phone number for
        ↪ general inquiries.",
    "steps": [
        "Navigate to 'boldtcastle.com'.",
        "Click on the 'Visiting' link.",
        "Click on the 'Plan Your Visit' link.",
        "Identify the 2025 season operating dates and hours for
            ↪ Boldt Castle.",
        "Find the Boldt Castle-only admission price for an adult
            ↪ (13+ years).",
        "Find the Boldt Castle-only admission price for a child aged
            ↪  6 (5-12 years).",
        "Locate information on how to get to Boldt Castle on Heart
            ↪ Island.",
        "Find the general inquiry phone number for Boldt Castle.",
        "Synthesize all collected information into a concise answer
            ↪ ."
    ],
    "criteria": [
        "State the 2025 season operating dates and hours for Boldt
            ↪ Castle as May 10 - October 13, 10:30 AM - 6:30 PM.",
        "State the adult admission price for Boldt Castle only as
            ↪ $13.50.",
        "State the admission price for a child aged 6 for Boldt
            ↪ Castle only as $9.50.",
        "Provide the physical location of Boldt Castle (Heart Island
            ↪ , Alexandria Bay, New York) and mention it's only
            ↪ accessible by water.",
        "Provide the general inquiry phone number for Boldt Castle
            ↪ as 315-482-9724."
    ]
}
```

### `visitwestchesterny.com`

In this example, we explored `visitwestchesterny.com` and saw a
    ↪ page that lists various coffee houses in Westchester County,
```

```
          ↪  including their names, addresses, and links to their
          ↪ descriptions.

 ```json
 {
     "proposed_task": "Find a cozy coffee shop in Westchester County
         ↪ . Navigate to the 'Coffee Houses' section on the Visit
         ↪ Westchester NY website. Find a coffee shop described as '
         ↪ cozy' and provide its name, full address, and the exact
         ↪ sentence from its description that indicates it is cozy
         ↪ .",
     "steps": [
         "Navigate to 'visitwestchesterny.com'",
         "Click on 'Things to Do'",
         "Click on 'Food and Drink'",
         "Click on 'Coffee Houses'",
         "Scroll down to view the coffee shop listings.",
         "Identify 'Altamira Cafe Bar' (or any other coffee shop)
             ↪ described as 'cozy'.",
         "Extract the name and address of the identified coffee shop
             ↪ .",
         "Click the 'Details' link for the identified coffee shop.",
         "Identify and extract the exact sentence from the
             ↪ description on its dedicated page that indicates it is
             ↪  cozy."
     ],
     "criteria": [
         "Successfully navigate to the 'Coffee Houses' page.",
         "Identify a coffee shop described as 'cozy' (e.g., 'Altamira
             ↪  Cafe Bar').",
         "State the name of the identified coffee shop (e.g., '
             ↪ Altamira Cafe Bar').",
         "State the full address of the identified coffee shop (e.g.,
             ↪ '245 Main St., New Rochelle, NY 10801').",
         "Successfully navigate to the 'Details' page for the
             ↪ identified coffee shop.",
         "Correctly state the exact sentence from the description
             ↪ that indicates it is cozy (e.g., 'Relax in the cozy
             ↪ shop or take a treat to go with piping hot espresso, a
             ↪  cold coffee, delicious desserts and delightful
             ↪ sandwiches.')."
     ]
 }
 ```
```

### `odetterestaurant.com`

In this example, we explored `odetterestaurant.com` and saw a '
    ↪ Reservations' page, which lists policies for dietary
    ↪ accommodations, birthdays, a deposit requirement,
    ↪ cancellations, and rescheduling.

```json
{
    "proposed_task": "I want to make a dinner reservation for 4
        ↪ people at Odette, and one of my guests has a severe dairy
        ↪  allergy. I also want to request a birthday cake for the
        ↪ table. What are the key policies I need to be aware of
```

```
          ↪ regarding my guest's allergy, the cake request, and any
          ↪ deposit or cancellation rules for this reservation?",
      "steps": [
         "Navigate to 'odetterestaurant.com'",
         "Go to the 'Reservations' page",
         "Identify the policy regarding dairy allergies and other
             ↪ dietary accommodations",
         "Find the policy for requesting a birthday cake, including
             ↪ notice period and cost",
         "Locate the deposit requirement per person for dinner
             ↪ reservations",
         "Determine the cancellation or rescheduling policy and
             ↪ associated timeframe",
         "Synthesize all relevant policies into a concise answer"
      ],
      "criteria": [
         "State that Odette is unable to accommodate guests with
             ↪ dairy allergies or intolerance.",
         "State that cakes require a 72-hour notice and cost SGD78
             ↪ ++.",
         "Confirm a deposit of SGD200 per person is required for
             ↪ dinner reservations.",
         "State that all reservations are final and non-refundable,
             ↪ but changes can be made at least 72 hours prior to the
             ↪  reservation date."
      ]
}
```

### `dottyabouticecream.co.uk`

In this example, we explored `dottyabouticecream.co.uk` and saw a
    ↪ form for hiring Dotty's ice cream van for corporate events,
    ↪ which includes fields for event details, guest count, and
    ↪ flavor inquiries.

```json
{
   "proposed_task": "Inquire about hiring Dotty's ice cream van
       ↪ for a corporate event in Manchester, M1 1AE, on August 15
       ↪ th, 2024, from 2 PM to 4 PM, for 100 guests. Ask if
       ↪ vanilla, honeycomb crunch, and mango sorbet are available.
       ↪  Fill out the 'Get in Touch' form with your details (Jane
       ↪  Doe, jane.doe@example.com, 07123456789) and note you
       ↪ found them via a web search. Do not submit the form.",
   "steps": [
      "Navigate to the 'Get in Touch' page on dottyabouticecream.
          ↪ co.uk.",
      "Fill 'Jane Doe' into the 'Name' field.",
      "Fill 'jane.doe@example.com' into the 'Email' field.",
      "Fill '07123456789' into the 'Telephone Number' field.",
      "Fill 'August 15th, 2024' into the 'Event Date' field.",
      "Fill '2 PM - 4 PM' into the 'Time of Ice Cream Service'
          ↪ field.",
      "Fill 'Manchester, M1 1AE' into the 'Venue Address (incl.
          ↪ Postcode)' field.",
      "Fill '100' into the 'Number of Expected Guests' field.",
      "Fill 'Web Search' into the 'Where Did You Hear About Dotty
          ↪ ?' field.",
```

```
            "Fill the 'Message' field with an inquiry about the
                ↪ availability of 'Vanilla, Honeycomb Crunch, and Mango
                ↪ Sorbet' flavors for a corporate event.",
            "Confirm all specified fields are accurately filled, but do
                ↪ not click the 'Gimmie Ice Cream' submit button."
        ],
        "criteria": [
            "The agent successfully navigates to the 'Get in Touch' page
                ↪ .",
            "The 'Name' field is filled with 'Jane Doe'.",
            "The 'Email' field is filled with 'jane.doe@example.com'.",
            "The 'Telephone Number' field is filled with
                ↪ '07123456789'.",
            "The 'Event Date' field is filled with 'August 15th,
                ↪ 2024'.",
            "The 'Time of Ice Cream Service' field is filled with '2 PM
                ↪ - 4 PM'.",
            "The 'Venue Address (incl. Postcode)' field is filled with '
                ↪ Manchester, M1 1AE'.",
            "The 'Number of Expected Guests' field is filled with
                ↪ '100'.",
            "The 'Where Did You Hear About Dotty?' field is filled with
                ↪ 'Web Search'.",
            "The 'Message' field clearly inquires about the availability
                ↪  of 'Vanilla, Honeycomb Crunch, and Mango Sorbet'
                ↪ flavors for a corporate event.",
            "The agent does not submit the form by clicking the 'Gimmie
                ↪ Ice Cream' button."
        ]
}
```

### `engineered.polestar.com`

In this example, we explored `engineered.polestar.com` and saw a
    ↪ page with information about Polestar Engineered Optimization
    ↪  for various Volvo models, including the 2023 Volvo XC60
    ↪ with a B5 Drive-E engine.

```json
{
    "proposed_task": "Is a Polestar Engineered Optimization
        ↪ available for a 2023 Volvo XC60 with a B5 Drive-E engine?
        ↪  If so, what are the primary performance benefits, how
        ↪ long does the installation take, and how would I find a
        ↪ dealer for this service?",
    "steps": [
        "Navigate to engineered.polestar.com.",
        "Under 'Can My Volvo Be Optimised?', select 'XC' then 'New
            ↪ XC60' for the model.",
        "Locate and select 'XC60 B5 Drive-E AWD Automatic 2023' or '
            ↪ XC60 B5 Drive-E FWD Automatic 2023' to view its
            ↪ optimization details.",
        "Confirm if the vehicle is 'Approved for Polestar Engineered
            ↪  Optimization'.",
        "Identify the primary performance benefits listed for the
            ↪ optimization.",
        "Determine the approximate installation time.",
```

```
            "Click on any 'Find a retailer' or 'Contact a dealer' links
                ↪ to see where they lead.",
            "Based on the website's information, describe how a user
                ↪ would find a dealer for installation.",
            "Synthesize all gathered information to answer the task."
        ],
        "criteria": [
            "Confirm that a 2023 Volvo XC60 with a B5 Drive-E engine is
                ↪ 'Approved for Polestar Engineered Optimization'.",
            "State the primary performance benefit as 'Power Mid-Range
                ↪ up to (hp) +3%' (from the specific product page) or '
                ↪ Up to +15% increased mid-range power' (from the
                ↪ general 'Get optimisation' page).",
            "State that the installation takes 'less than 60 minutes'.",
            "Clearly state that clicking the 'Find a retailer' or '
                ↪ Contact a dealer' links does not lead to a functional
                ↪ dealer search tool, and that users are advised to
                ↪ contact their local Volvo retailer directly for
                ↪ further questions."
        ]
}
```

### `ajga.org`

In this example, we explored `ajga.org` and saw a page with
    ↪ information about Performance Based Entry (PBE) Stars for
    ↪ junior golfers, including how they carry over to the next
    ↪ season and tips for maximizing tournament opportunities.

```json
{
    "proposed_task": "As a junior golfer, help me understand how my
        ↪ Performance Based Entry (PBE) Stars carry over to the
        ↪ next season on the AJGA circuit. Additionally, I'm
        ↪ looking for two tips from the website on how to best
        ↪ maximize my opportunities to play in tournaments. Can you
        ↪ provide this information?",
    "steps": [
        "Navigate to the 'Juniors' section of the AJGA website.",
        "Click on 'How to Play in the AJGA'.",
        "Click on the 'PBE' link within the 'How to Play in the AJGA
            ↪ ' section.",
        "Scroll down to locate and click on the 'Important Notes and
            ↪ Tips' link.",
        "Identify and extract information regarding the carry-over
            ↪ of Performance Stars to the next season.",
        "Identify and extract two distinct tips that help players
            ↪ maximize their tournament playing opportunities.",
        "Synthesize the gathered information to provide a
            ↪ comprehensive answer."
    ],
    "criteria": [
        "Explain that Performance Stars earned in one year (e.g.,
            ↪ 2024) are carried over to the beginning of the next
            ↪ season (e.g., 2025) and are combined with new
            ↪ membership Performance Stars based on grad year.",
```

```
          "Identify and state that 'Plan your tournament schedule
              ↪ early to maximize playing opportunities and prevent
              ↪ missed deadlines.' is a key recommendation.",
          "Identify and state that 'Qualifiers are great opportunities
              ↪  for all players to earn Performance Stars and build
              ↪ their status.' is a second key recommendation."
      ]
}
```

### `passports.gov.au`

In this example, we explored `passports.gov.au` and saw a page
    ↪ with a section about the documents needed to prove
    ↪ Australian citizenship for individuals born in Australia on
    ↪ or after August 20, 1986.

```json
{
    "proposed_task": "I was born in Australia on or after August
        ↪ 20, 1986, and am applying for my first Australian
        ↪ passport. What documents do I need to prove my Australian
        ↪  citizenship? Please list the primary document,
        ↪ acceptable alternatives, and any specific requirements
        ↪ for proving citizenship by birth based on my parents' or
        ↪ grandparents' status.",
    "steps": [
        "Navigate to 'passports.gov.au'.",
        "Navigate to the 'How it works' section.",
        "From 'How it works', navigate to 'Documents you need'.",
        "On the 'Documents you need' page, navigate to the '
            ↪ Citizenship' section.",
        "Within the 'Citizenship' section, locate the information
            ↪ for individuals 'Born in Australia on or after 20
            ↪ August 1986'.",
        "Identify the primary document required for proof of
            ↪ citizenship.",
        "Identify and list all acceptable alternative documents.",
        "Detail the specific scenarios for proving citizenship by
            ↪ birth, including those involving parents' or
            ↪ grandparents' documentation and the special case for
            ↪ permanent resident parents."
    ],
    "criteria": [
        "State that the primary document is the applicant's full
            ↪ Australian birth certificate.",
        "List an Australian citizenship certificate in the applicant
            ↪ 's name as an acceptable alternative.",
        "List an Australian passport issued in the applicant's name
            ↪ on or after 1 January 2000 that was valid for at least
            ↪  two years as an acceptable alternative.",
        "Detail the scenario where one parent was an Australian
            ↪ permanent resident or citizen, specifying the required
            ↪  parental documents (birth certificate, passport, or
            ↪ citizenship certificate).",
        "Explicitly mention that if both parents were Australian
            ↪ permanent residents when the applicant was born,
            ↪ evidence of citizenship must be obtained from the
            ↪ Department of Home Affairs.",
```

```
          "Include the scenario involving grandparents' documents (
              ↪ birth certificate, passport, or citizenship
              ↪ certificate) if the parent was born in Australia on or
              ↪  after 20 August 1986."
      ]
}
```

## Formatting Your Response

Establish how the task can be refined in at most 300 words, and
    ↪ synthesize relevant content and features on the website in
    ↪ your response. After your response, provide a refined task
    ↪ as JSON in a fenced code block.
```

## H   DETAILS FOR TRAINING AGENTS

To understand the utility of data we obtained, we train agents and test on four relevant benchmarks: InSTA, WebVoyager (He et al., 2024), Mind2Web (Deng et al., 2023), WebLINX (Lù et al., 2024). In particular, our test set consists of a held-out set of 3,000 websites and tasks produced by the task generation feedback loop. Note these websites are not present in the training set. For WebVoyager (He et al., 2024), we transfer agents trained on our data zero-shot to 643 tasks on 15 websites WebVoyager (He et al., 2024). The websites in the WebVoyager benchmark are not present in the 20k trajectories we collected in Section 6.1 we used for training agents. For this experiment, we fine-tuned models based on *Qwen 3 1.7B* with a maximum sequence length of 16,384 tokens, and the most recent 5 observations, and actions in the context. We employed full fine-tuning on this model, with Adam, learning rate of `5e-5`, batch size of 32, `bfloat16`, and other parameters kept as the PyTorch defaults for Adam. Each model was trained using one epoch, a linear warm-up corresponding to the first 0.01 steps of training, and a linear decay to `6e-5` afterwards.

To filter data, we select trajectories that were scores as `Judge(Success) = 1`, which corresponded to 10.5k of 20k trajectories produced by the *Qwen 3 235B* data collection policy. Scores for filtering were produced by a *Qwen 3 235B* judge. We employed a simple filtering strategy that only considers the success score from the judge, and no other filtering conditions were used. Note the judge also predicts efficiency, and self correction scores, which could likely also help select the best data for training, but we did not explore filtering by these scores in this work.

For experiments on static benchmarks, we fine-tune `google/flan-t5-large` for Mind2Web, and `meta-llama/Llama-3.1-8B-Instruct` for WebLINX using official fine-tuning code released with corresponding benchmarks. We employ identical training hyperparameters to those used by Lù et al. (2024) for Llama in their official training code and Deng et al. (2023) for Flan to ensure that our results are comparable to previous work. Section 6.2 reports performance on the official `test_web` split of the WebLINX benchmark, and the official `test_website` split of the Mind2Web benchmark, where agents are tested on unobserved websites. The websites in these static benchmarks were not present in the dataset we generated for this experiment to ensure fairness.

## I   HYPERPARAMETERS

We provide a list of the hyperparameters used in this work in Table 3. Hyperparameters for mixing our data with human data on static benchmarks are selected to mirror prior work in synthetic data (Trabucco et al., 2024), and to adhere to standard hyperparameters for WebLINX (Lù et al., 2024), and Mind2Web (Deng et al., 2023). We train using all available human data on these benchmarks, and add trajectories filtered using the previously discussed methodology, sampled at a 20% rate in the data-loader compared to an 80% rate for human data.

| Hyperparameter Name | Value |
|---|---|
| Models Used For Agents | `Qwen/Qwen3-1.7B` |
| | `Qwen/Qwen3-235B-A22B` |
| | `meta-llama/Llama-4-Maverick-17B-128E-Instruct` |
| | `meta-llama/Llama-3.1-70B-Instruct` |
| | `meta-llama/Llama-3.3-70B-Instruct` |
| | `google/gemini-2.5-flash` |
| Models Used For Judges | `Qwen/Qwen3-235B-A22B` |
| | `meta-llama/Llama-4-Maverick-17B-128E-Instruct` |
| | `meta-llama/Llama-3.1-70B-Instruct` |
| | `meta-llama/Llama-3.3-70B-Instruct` |
| | `google/gemini-2.5-flash` |
| | `google/gemini-1.5-pro` |
| | `openai/gpt-4.1-mini` |
| | `openai/gpt-4o` |
| Common Crawl PageRank | `cc-main-2024-apr-may-jun-host-ranks.txt.gz` |
| Number of sites before filtering | $1,000,000$ |
| Number of tasks after filtering | $146,746$ |
| Max Tokens Per Observation | $2,048$ |
| Max Tokens Per Agent Trace | $1,024$ |
| Max Tokens Per Judge Trace | $1,024$ |
| Max Tokens Per Task Proposer Trace | $1,024$ |
| Last Steps Per Agent Context | $5$ |
| Last Steps Per Judge Context | $5$ |
| Last Steps Per Task Proposer Context | $5$ |
| Task Proposer Feedback Loops | $1$ |
| LLM Sampling Temperature | $0.5$ |
| LLM Sampling Top P | $1.0$ |
| LLM Sampling Top K | default |
| Fine-tuned LLM in Section 6.1 | `Qwen/Qwen3-1.7B` |
| InSTA Training Epochs | $1$ |
| InSTA Batch Size | $32$ |
| InSTA Learning Rate | `5e-5` |
| InSTA Optimizer | Adam |
| Mind2Web LLM | `google/flan-t5-large` |
| Mind2Web Training Iterations | $11,505$ |
| Mind2Web Batch Size | $32$ |
| Mind2Web Learning Rate | `5e-5` |
| Mind2Web Optimizer | Adam |
| WebLINX LLM | `meta-llama/Llama-3.1-8B-Instruct` |
| WebLINX Training Iterations | $10,000$ |
| WebLINX Batch Size | $16$ |
| WebLINX Learning Rate | `5e-5` |
| WebLINX Optimizer | Adam |
| Data Filtering Condition | `Judge(Success) = 1` |
| Human Data Sampling Probability $p_{\text{real}}$ | $80\%$ |

Table 3: **Hyperparameters used in our paper.** We organize hyperparameters into seven sections, for the names of LLMs used as agents in the paper, the names of LLMs used as judges in the paper, the hyperparameters used for data collection, the sampling parameters for LLMs, the training parameters for static benchmarks, and the filtering and data mixing hyperparameters.

## J   COST ANALYSIS FOR LLAMA 3.1 70B

To understand the significant reduction in cost that we obtain by running LLMs locally to generate data, we analyze the number of tokens processed by the LLM, and compute an expected cost if this were served using proprietary models. As the analysis shows, using *Llama 3.1 70B* is a feasible

option for running agents at this large scale, and results in the paper show that this choice of LLM backbone does not compromise performance. We have deep gratitude to the Llama team at Meta, and the Qwen team at Alibaba for working to make developments in language modeling available to the research community at no cost. We see up to a 95% reduction in cost with these models.

| Variable Name | Value |
|---|---|
| Number of tasks | $146,746$ |
| Average tokens per observation | $1,024$ |
| Max observations per agent context window | 5 |
| Average agent / judge response size | 512 |
| Max tokens per system prompt | $1,024$ |
| Average steps per task | 15 |
| Estimated tokens processed by the agent | 14.65B tokens |
| Tokens processed by the judge | 1.35B tokens |
| Total tokens processed | 16.00B tokens |
| Expected API cost for *GPT-4.1* | $ 32,000.00 |
| Expected API cost for *Gemini 2.5 Pro* | $ 20,000.00 |
| Expected AWS compute cost for serving *Llama 3.1 70B* (3,840 v100 GPU hours using spot instances) | $ 1,575.70 |
| Percent saved using *Llama 3.1 70B* | $[95.08, 92.12]$ % |

Table 4: **Cost analysis for different LLM models in the fully-scaled pipeline.** This table provides statistics for the number of tokens that were processed by our pipeline, and why serving using a local LLM engine like vLLM is important for bringing down costs.

