# OpenReview forum: "InSTA: Towards Internet-Scale Training For Agents"
_ICLR.cc/2026/Conference — Submitted to ICLR 2026_

### Official Review · Reviewer_vZ2x · 2025-10-30

**Soundness:** 2
**Presentation:** 3
**Contribution:** 2
**Rating:** 4
**Confidence:** 4

**Summary:**

The paper proposes a 3-stage, LLM-driven pipeline for creating training data for web agents. An LLM task proposer takes websites from the top 1 million in Common Crawl, ensures the domain is safe, and writes a task for the site. An agent then generates trajectories based on these tasks, and the proposer generates new, more difficult tasks based on these trajectories. An LLM judge scores the trajectories. A small agent is then trained using this generated and judged data, and performs comparably to much larger models (on their data). They claim zero-shot transfer to WebVoyager.

**Strengths:**

- The scale is large enough that the data is likely to be useful to others.
- Train/test separation of websites is stricter than in many other papers.
- Transfer to WebVoyager slightly mitigates overfitting concerns.

**Weaknesses:**

- Claims about the accuracy of safety filtering and trajectory judging are based on samples of 100 websites and 100 trajectories, respectively. I believe this is too small to make this claim, and unlikely to be representative of your 150k site sample.
- Headline numbers are based on LLM-as-judge evaluation, without any human verification/sanity checking (beyond that detailed above).
- The WebLinx and Mind2Web experiments do not demonstrate transfer to existing benchmarks. There is no improvement on their test sets, only on yours, which seems entirely expected.
- The only external data on which we see improvement is WebVoyager, which is again LLM judged. This does not rule out potential reward hacking.

**Questions:**

- Can you report human-evaluated success on a larger subset of your data, and show that the 1.7B model remains competitive? Alternatively, could you run your agent and compare it with the other models on static, rule-based benchmarks such as WebArena (Zhou, 2023) and others? Either of these would reduce my concern about reward hacking and provide stronger evidence that the method is viable.
- What is the transfer to Mind2Web and WebLinx without using their human demonstration data?

---

### Official Review · Reviewer_36Q5 · 2025-11-01

**Soundness:** 2
**Presentation:** 3
**Contribution:** 2
**Rating:** 4
**Confidence:** 4

**Summary:**

This paper proposes a pipeline for collecting web trajectory data. The corresponding dataset is also released. The trajectories are generated by starting with an initial task, followed by exploration on the website, and then generating the task description based on the grounded trajectory. The authors make efforts to filter harmful websites. Trajectory evaluation is done using LLM as a judge. To evaluate the usefulness of trajectories, they evaluate on WebVoyager, WebLINX, and Mind2Web. Further, they show that performance improves with data scale.

**Strengths:**

1. The trajectory dataset is a valuable resource.
2. The performance improvement with scale is a positive indicator of dataset quality.

**Weaknesses:**

1. Missing external baselines for Mind2Web and WebLINX, and WebVoyager. Also, it is not clear how the set of 500 diverse test tasks for Mind2web was chosen.
2. Missing references - Explorer [1] - similar pipeline for web agent trajectory synthesis

[1] Pahuja, Vardaan, et al. "Explorer: Scaling exploration-driven web trajectory synthesis for multimodal web agents." ACL 2025.

**Questions:**

1. What is the evaluation benchmark used in section 6.1? Is it an intrinsic evaluation on the test set of generated trajectories? Please elaborate.

---

### Official Review · Reviewer_4wrC · 2025-11-01

**Soundness:** 3
**Presentation:** 4
**Contribution:** 4
**Rating:** 10
**Confidence:** 3

**Summary:**

This paper proposes a method for training web navigation agents based on (1) selecting a subset of 150K websites from Common Crawl, (2) generating sample tasks on each of these websites, and (3) training a filtered BC model based on the outputs of an LLM judge. The authors run this recipe with a Qwen-3 235B model and distill it into a Qwen-3 1.7B model, outperforming the original 235B model.

**Strengths:**

1. This is a great paper, with really impressive results, showing that frontier agent performance can be achieved with just a few hundred dollars. I expect it  will be of broad interest to agent researchers

2. The task generation setup is clever — for each website, the proposed method first generates a simple task which gives an agent incentive to explore the website, and then generates more complex tasks conditioned on this trajectory.

3. The paper conducts two useful human validations: first, to make sure that the majority of LLM-proposed tasks are in fact achievable by humans, and second to confirm that the LLM judge models produce mostly correct judgments

4. This paper opens up many promising lines of future work, such as scaling the current recipe, experimenting with new approaches to task generation, and expanding to RL methods beyond filtered BC

**Weaknesses:**

I think the abstract/intro could have been a bit more clear that the headline result requires both the policy and judge models to be 235B models, but given that the final performance of the distilled 1.7B model surpasses the performance of the 235B model, I don’t consider this to be a major weakness.

**Questions:**

N/A

---

### Official Review · Reviewer_Esoq · 2025-11-02

**Soundness:** 3
**Presentation:** 3
**Contribution:** 2
**Rating:** 4
**Confidence:** 3

**Summary:**

This paper proposes InSTA, an automatic, three-stage pipeline to facilitate internet-scale agent trajectory collection and training. The first stage is to use an LLM-based task proposer to filter 1 million websites for safety, and generate agentic tasks for 150,000 diverse sites. This stage includes a feedback loop where an agent's trajectory is used to generate a harder, more grounded task. In the second stage, an LLM agent (i.e. the data collection policy) is employed to attempt the generated tasks on live websites, producing trajectories of observations and actions. Finally, an LLM-based judge is used to evaluate the success of these trajectories, filtering for high-quality data.

Using this pipeline, the authors create a large-scale dataset including 2.2M steps. Their key finding is that a small model (Qwen 3 1.7B) trained on this judge-filtered data can outperform much larger, zero-shot frontier models Llama 4 Maverick and Qwen 3 235B. The trained agents also demonstrate strong zero-shot transfer to the Web Voyager benchmark and improve the generalization of agents trained on static benchmarks like Mind2Web and WebLINX.

**Strengths:**

1. The pipeline's execution at the scale of 150,000 live websites is a non-trivial engineering and research achievement, far surpassing the ~200 sites in many existing benchmarks. The release of the insta-150k-v2 task dataset and the larger reasoning dataset is a valuable contribution to the community.
2. The authors prudently integrate safety considerations from the start, rather than treating it as an afterthought. The LLM-based safety filter is shown to be highly effective (up to 97% accuracy) , and the tasks generated are validated by humans as being achievable and verifiable (up to 89%).

**Weaknesses:**

1. The feedback loop is proposed as a key design in section 4.1. However, the paper explicitly states, "For this paper, we employ one loop of task generation". This is a major limitation, which indicates that the full promise of an iterative system where tasks get incrementally harder is not actually realized or evaluated.
2. The paper claims to generate "challenging" tasks. However, the analysis in Appendix F (Figure 14) shows the most solved tasks are simple information retrieval (e.g., "contact information," "hours of operation," "biographical info"). Conversely, Figure 15 shows that the least successful categories are more complex tasks like "product comparison," "downloading reports," and "booking". This suggests the InSTA pipeline is currently most effective at creating simple lookup tasks, not the complex, multi-step reasoning and interaction tasks that are a key frontier for agent research.

**Questions:**

The paper's feedback loop is a core idea, but the experiments only use one loop. Can the authors provide results using multiple loops of task generation? Is there quantitative evidence that tasks from $loop_{n+1}$ are measurably "harder" or lead to better agent training than tasks from $loop_n$? Without this, the dynamic environment claim seems premature.

---

### Meta-Review · Area_Chair_oNV3 · 2026-01-11

**Summary:**

Reviewers agree that the paper presents an ambitious pipeline for generating web agent tasks and trajectories using LLMs, and acknowledge the engineering effort involved in operating at the scale of 150k websites **[Esoq, 4wrC, 36Q5, vZ2x]**.
One reviewer views the results as particularly strong, highlighting that a small distilled model achieves competitive performance with much larger frontier models **[4wrC]**.

However, multiple reviewers raise substantive concerns about the strength of the paper’s claims and the rigor of its evaluation.
Reviewers question whether the proposed feedback loop meaningfully increases task difficulty with a single round iteration **[Esoq]**.
They also observe that most generated tasks are simple information retrieval, while more complex multi-step interactions remain largely unsolved **[Esoq]**.
Several reviewers point out missing or weak external baselines on established benchmarks such as Mind2Web, WebLINX, and WebVoyager **[36Q5, vZ2x]**.
Finally, reviewers raise concerns about the heavy reliance on LLM-as-a-judge evaluation with limited human validation, which leaves open the possibility of reward hacking and unreliable success estimates **[vZ2x]**.

The authors did not participate in the rebuttal, leaving these concerns unaddressed. Overall, reviewer sentiment remains mixed, with the majority of reviewers below the acceptance threshold.

**Reviewer Concerns:**

**Addressed**
- None. The authors did not participate in rebuttal

**Outstanding**
- Effectiveness of the feedback loop: Esoq questions whether using only a single loop supports claims about progressively harder task generation ([Esoq]).
- Task difficulty and diversity: Esoq notes that most successful tasks are simple lookups, while more complex tasks remain largely unsolved ([Esoq])
- External baselines and evaluation clairty: 36Q5 and vZ2x point out missing baselines and unclear evaluation protocols for Mind2Web, WebLINX, and WebVoyager ([36Q5, vZ2x]).
- Reliance on LLM-as-a-judge: vZ2x raises concerns about reward hacking and limited human validation given small audit samples ([vZ2x]).

**Reviewer Scores:**

Again, the authors did not participate in the rebuttal or discussion.

---

### Decision · Program_Chairs · 2026-01-26

Reject